# Quality of Frozen Hake Fillets in the Portuguese Retail Market: A Case Study of Inadequate Practices in the European Frozen White Fish Market

**DOI:** 10.3390/foods10040848

**Published:** 2021-04-13

**Authors:** Rogério Mendes, Helena Silva, Patrícia Oliveira, Luís Oliveira, Bárbara Teixeira

**Affiliations:** Department for the Sea and Marine Resources, Portuguese Institute for the Sea and Atmosphere—IPMA, I.P., R. Alfredo Magalhães Ramalho, 6, 1449-006 Lisbon, Portugal; hsilva@ipma.pt (H.S.); pioliveira@ipma.pt (P.O.); luis.oliveira@ipma.pt (L.O.); barbara.teixeira@ipma.pt (B.T.)

**Keywords:** seafood quality, market study, adulteration, labeling, consumer’s evaluation, sensory analysis

## Abstract

The overall quality of frozen hake fillets in the Portuguese market was evaluated. Physical, biochemical, microbiological, and sensory analysis in 20 brands revealed several non-conformities. Hake was identified in 19 brands, although two mislabeled the species. Lower net weight than labeled was evidenced in ca. one-third of brands. TVB-N in ca. one-third of the samples presented high values, although within legal limit. Almost all brands presented excessive amount of ice glaze, low levels of WHC (raw/cooked), low levels of soluble protein, and undue thaw-drip loss, thus reflecting the poor quality of fillets. Added phosphates were below the legal limit in all samples; however, they were used in glaze ice in three brands, and only labeled in one. Overall microbiological quality of frozen fillets was good, though yeasts and molds detected in six samples indicate poor hygienic conditions in some facilities. Labels comply with legal requisites, however, need improvement, namely the ‘best before’ periods. In general, packaging was efficient regarding presence of ice crystals and dehydration signs. Concerning sensory analysis of cooked fillets, 63% of the brands had bad to poor quality and 37% fair quality. *Merluccius productus* revealed the worst quality, namely regarding thaw drip loss, thaw drip loss protein, M/P ratio, pH and sensory evaluation. Overall results show that a significant part of the hake fillets business operators are still failing in relation with best practices.

## 1. Introduction

Fish in the frozen form represented, in 2016 in Europe and all other developed countries, 58% of the total production used for human consumption [1]. Of all finfish landed in Europe in 2017, hake had the highest value, with European hake (*Merluccius merluccius*) representing 76% of the total hake. Together with the other 11 species that share this commercial designation, hake is one of the most traded and consumed products in Europe and was responsible in 2016 for a 4% share of the total seafood consumption [2]. Regarding European imports in the same year, hake had a share of 15%, making it the third-most imported finfish, behind cod and Alaska Pollock [3].

With an annual average consumption of 55.3 kg of fish per capita, Portugal is the third largest seafood consumer in the world among the developed countries and the biggest per capita fish consumer between European countries [4,5,6]. Portuguese seafood supply is highly dependent of global markets, with almost two-thirds of Portuguese seafood being imported [7]. On the other hand, chilled and frozen seafood were reported to have parallel relevance in the diet of the Portuguese population [5]. Hake is the third most consumed fish species (with 6%) by the Portuguese, after salt and dried cod (38%) and canned tuna (7%) and its significance has been linked to the fact that frozen hake products became very popular in the beginning of the 20th century [7]. Hake is rich in proteins of high biological value and low-fat content with mostly unsaturated fatty acids [8].

Currently available freezing techniques and industrial standard operating procedures involving processing at sea and quick-freezing techniques in fish processing ships allow the production of high-quality frozen seafood products [9]. Although freezing of meat is an effective method of long-term preservation, fish and fishery products can suffer undesirable changes during frozen storage (e.g., protein denaturation and aggregation, enzymatic hydrolysis of lipids, and proteins). In this regard, quality issues in frozen fish products relate mostly with dehydration and oxidation, and deterioration limits the storage time [10,11]. 

Alterations during frozen storage result in modification of the functional properties of the proteins, decrease of the water holding capacity (WHC) and deterioration of color and flavor, together with undesirable changes in texture, which causes hard, dry and fibrous products [12,13]. Changes are particularly relevant after extended periods of storage, when the initial fish quality is low and when poor freezing practices or high storage temperatures are employed. Texture and functional properties alterations are the result of denaturation and aggregation of muscle proteins, particularly those in the myofibrillar fraction [14], and are especially relevant processes in some lean species, like hake [15,16]. 

When myofibril protein denaturation or aggregation occurs during frozen storage, the WHC decreases and muscle tissue does not completely restore during thawing, leading to a significant deterioration of quality, because moisture that is not reabsorbed flows out as drip loss [17]. Processes occurring during frozen storage, like protein surface dehydration or, to a minor degree, physical damage in the myocytes, have been considered the causative processes of this decrease [16]. Likewise, texture, flaking, juiciness, and tenderness are also related to the WHC since all depend on heat denaturation of proteins. On account of this, WHC has been used in quality evaluation [18] not only for describing in raw muscle the ability of raw material to retain water during centrifugation, but also in cooked muscle to determine the effect of heat denaturation of proteins. Induced by denaturation/aggregation of proteins and quality deterioration throughout frozen storage, cook loss occurs also during the heat processing of hake fillets. During cooking of fish muscle, pressure gradients are formed because of protein network shrinkage due to denaturation, and in their presence, excess liquid consisting of water, dissolved proteins, ash, salt, and fat is moved to the surface of the fish [19].

Other variations like muscle pH also occur in seafood, generally because of biochemical and/or microbiological changes [20]. Small increases in pH in the order of 0.1–0.4 units have been accounted as resulting from the production of basic compounds in muscle tissue, namely ammonia [21]. Though pH levels are poor freshness indicators due to considerable variation between species, individual specimens and biological variation, a pH value of 7.0 can in most tests be considered as an indication of spoilage [22]. Similarly, pH 7.0 was also recommended as the maximum limit for rejection of frozen fish of the *Merlucciidae* family, though not to be used as a single parameter to judge seafood quality or even to consider it as unfit for consumption [21]. Generated also by endogenous enzymatic reactions, lipid autoxidation, and/or microbial spoilage actions particularly during storage temperatures above 0 °C [23], volatile amines (e.g., TVB-N) are the most characteristic molecules responsible for odor and taste present in fish, though levels change at variable rates depending on numerous factors (e.g., fish species, storage temperature).

On account of the quality loss during frozen storage, ‘best before’ dates or the period in which seafood can effectively be stored without creating undue quality damage to the product (practical storage life), are mandatory in the EU in what concerns frozen seafood labeling [24]. Since products quality can be affected by several factors (e.g., length of storage, temperature, ice glaze, type of packaging), definition of best before length is not regulated and processors will consider these factors when determining the date for which the product will be of best quality. Based on scientific data, several recommendations have been published as reference regarding the practical storage life of lean fish at −18 °C, namely 8 months [25], 9 months [9], and 8–10 months [26]. 

The legal framework addressing in Europe the quality of frozen products is diverse and involves several quality parameters, though not always sufficiently detailed. Health standards for fishery products are identified by Regulation (EC) no. 853/2004 [27] laying down the specific hygiene rules applicable to products on the market for human consumption. In this regard, food business operators must, for example, carry out organoleptic examinations of the fishery products and ensure that these comply with any freshness criteria. Within the biochemical parameters, total volatile basic nitrogen (TVB-N) is used as an index to assess the keeping quality and shelf life of unprocessed fishery products and legal requirements in Regulation (EC) 2074/2005 have been established for the limits of this indicator in products of the *Merluccidae* family [28]. 

In terms of microbiological safety, the Commission Regulation (EC) no. 2073/2005 on microbiological criteria for foodstuffs [29], states that these should not contain micro-organisms or their toxins or metabolites in quantities that present an unacceptable risk for human health, namely should not contain *Salmonella*. Other European microbiological criteria have not yet been established, but the ICMSF [30] proposed an aerobic mesophilic count limit of 7 log cfu/g for fish that is fit for human consumption and Sciortino and Ravikumar [31] proposed <10^2^ cfu/g as guideline for *Staphylococcus aureus* and <10 cfu/g for *Escherichia coli*. Also, in terms of food safety and concerning the additives used in the fish processing industry, condensed phosphates can be added to fillets (frozen and deep-frozen) individually or in combination for several technological processes, including to retain natural moisture, inhibit flavors and lipid oxidation, aid emulsification, and for cryoprotection [32]. However, high phosphates intakes may cause health problems, such as cardiovascular, renal, and a decrease in iron and copper absorption, among others [33,34]. On account of this, according to Regulation (EC) 1333/2008 on food additives [35] phosphates are regulated in terms of the maximum limit added (5 g P_2_O_5_.kg^−1^), whereas citric acid and citrates are allowed *quantum satis*.

Labeling of seafood products is on the other hand regulated mainly by Regulation (EU) no. 1379/2013 which sets out rules on the mandatory and voluntary information to be provided for prepacked and non-prepacked fishery and aquaculture products [36] and Regulation (EU) no. 1169/2011 on the provision of food information to consumers [24]. These regulations address relevant issues like, the net quantity of the food, the commercial and scientific name of the species, date of minimum durability (‘best-before’ date), the list and quantity of certain ingredients or categories of ingredients, among others. Conformity of net weight and ice glaze follows specific Portuguese regulation [37] laying down the marketing conditions of frozen, deep-frozen, and thawed fishery and aquaculture products intended for human consumption and addressing the determination of drained net weight and glaze water content of frozen and deep-frozen glazed products.

Besides using commercial packaging materials and wraps as effective barriers to minimize dehydration and oxidation the fish-freezing industry has made general use of ice glazes to reduce these same effects. However, quality is not always up to standards in what concerns for example the weight statement on pre-packaged frozen seafood, which must refer to the net weight of the seafood excluding any external water or ice glaze. Hake either in the chilled or frozen state, whole, in portions, or in fillets is one of the most appreciated seafood products in the Portuguese gastronomy and on account of that has often been the subject of quality evaluation studies by consumer’s associations [38,39]. Nevertheless, though the production and marketing of good quality frozen hake products can be warranted by the existence of adequate technological solutions the industry practice is not devoid of problems as some consumer studies have pointed out [38,39]. 

Hake species marketed and consumed in Portugal are mainly of the species *Merluccius merluccius*, *M. hubbsi*, *M. capensis*, *M. paradoxus*, and *M. productus*. Considering the number of traded hake species and the worldwide diversity of sources and products, frozen hake sold in Portugal is a good example of the global market practices for assessment of the level of compliance of operators regarding the European frozen white fish market. In that sense, the objective of the study is to evaluate the overall quality of frozen hake fillets in the Portuguese market as a case study for the characterization of the frozen white fish industry (processors and retailers), including the compliance with safety and labeling regulations and the consumer’s evaluation of fillets sensory quality. For that purpose, samples of deep-frozen hake fillets were purchase in the Portuguese retail market and sensory, physical, biochemical, and microbiology quality parameters analyzed. 

Selection of the biochemical analysis performed was based on the ability to better evidence the quality changes occurring during processing and frozen storage and also the compliance of products with labeling regulations. Other biochemical analysis namely in the lipid fraction, were not considered because hake is a lean species and most of the changes ascribed to freezing and frozen storage are caused by loss of textural attributes rather than oxidation. Following the approach of Bremmer [40] that linked “the concept of quality, through a general definition, by adding the missing link of specific definitions related to measurable attributes and properties determined by standard methods to provide values that can be used to evaluate foods or to set specifications”, a multivariate analysis of all data was performed to identify the most informative quality control parameters and also to detect groups of commercial hake samples with similar quality characteristics. 

## 2. Materials and Methods

### 2.1. Raw Material, Processing, and Sampling

#### 2.1.1. Market Study

In order to identify the number of brands of deep-frozen pre-packaged hake fillets available in the Portuguese retail market, a survey was carried out in 13 small traditional food shops (minimarkets, traditional markets and frozen fish shops) and large food retail chains (hyper/supermarkets). The survey detected a total of 32 hake fillet brands from which the 20 brands with the higher frequency of presence in the retailers were selected for analysis (Table 1). Given the growing importance of own brands of large food retail chains, about half of the sample is composed of these brands (45% large food retail chains brands and 55% manufacturer brands). A total of 13 packages of the same brand and batch were collected in October 2018 in the selected food shops, transported to the laboratory under controlled temperature (−18 °C) and kept stored at −20 °C ± 1 °C until analysis within 1 week. From these, 10 packages were used for analysis of packaging (packaging defects, presence of ice and fillet’s size variation), net drained weight, glaze ice, and sensory analysis and three packages (pooled) used in the other determinations. Individual weight of hake fillets packages ranged from 0.4 kg to 1.0 kg and no records of storage temperatures were obtained from processors or retailers.

#### 2.1.2. Chemicals and Reagents

All the chemicals and reagents used were analytical grade of the highest purity and supplied by Merck (Darmstadt, Germany). Aqueous solutions were prepared with ultra-pure Milli-Q (Merck Millipore, Billerica, MA, USA) purified water.

### 2.2. Analytical Determinations

#### 2.2.1. Physical Analysis

The determination of net drained weight and glaze ice percentage of deep-frozen hake fillets was carried out based on the method described on Codex Alimentarius [41] according to Portuguese Regulation [37]. Thaw drip loss was calculated as the weight percentage of water released by the fillets without ice glaze, following a 24 h thawing process in a refrigerated chamber at 4 °C ± 1 °C (Fiocchetti Labor 500 ECT-F, Luzzara, Italy). The water-holding capacity in raw (WHC_raw_) and cooked (WHC_cook_) fillets was measured with the method described by Sánchez-González et al. [42]. A portion of fillet (ca. 2 g; *W_s_*) and folded filter paper (also weighted, *W_i_*) were placed in the bottom of a centrifuge tube and centrifuged at 3000× *g* for 10 min at 20 °C (3K30, Sigma, Osterode, Germany). After centrifugation, the sample was removed and the filter paper was weighed (*W_f_*). WHC_raw_ and WHC_cook_ were expressed as g of water in sample after centrifugation per 100 g of water initially present in the sample:$$WHC=\frac{Ws\times \frac{M}{100}-\left(Wf-W1\right)}{Ws\times \frac{M}{100}}\times 100$$
where *M* is the moisture (g/100 g). All determinations were performed in duplicate. 

Cook loss was determined as the percentage in weight difference between the raw and cooked sample based on the raw weight. Analysis of cook loss was done in samples vacuum packed in 140 mm thickness polyamide and polyethylene film bags (Vaessen-Schoemaker, Ovar, Portugal) and steam-cooked in a pre-heated steam oven Rational Combi-Master model CM6 (Landsberg, Germany) for 15 min at 100 °C. All determinations were performed in triplicate.

#### 2.2.2. Biochemical Analysis

Moisture content was determined by the sample drying technique at 105 °C [43], whereas crude protein was determined in a model FP-528 DSP LECO protein/nitrogen analyser (LECO Corp., St. Joseph, MI, USA), according to the Dumas combustion method [44]. Total volatile basic nitrogen (TVB-N) was determined according to the modified micro diffusion method described by Cobb, Alaniz & Thompson [45]. The pH was measured directly on minced hake fillets using a calibrated pH insertion electrode (WTW Sentix sp; Weilhein, Germany) connected to a pH meter (WTW 7110 pH meter; Weilhein, Germany).

Determination of salt soluble protein content followed a method similar to that described by Woyewoda et al. [46]. Minced fillet (2.5 g) was homogenized with 25 mL of cold 5% NaCl solution buffered with 0.003 M NaHCO_3_ using a Polytron (Kinematica, A.G., Luzern, Switzerland) for 2 min (<5 °C). The homogenate was centrifuged (20,000× *g*, 20 min, 4 °C) and the nitrogen content in the supernatant analyzed by the Dumas combustion method. The results were expressed as g of protein per 100 g of sample, using 6.25 as a nitrogen to protein conversion factor. Thaw drip loss protein was calculated in the water released by the fillets subtracting the portion corresponding to glaze ice, following a 24 h thawing process at 4 °C ± 1 °C and the results were expressed as g of protein per 100 mL of free drip. 

To control the presence/absence of polyphosphates in seafood, and also for their quantification, different methodologies were used. Total phosphates were determined in the fillets by spectrophotometric quantification of the phosphorus content and conversion to phosphates, according to ISO 13730 method [47], using a UNICAM UV 5 220 spectrophotometer (ATI UNICAM, Cambridge, UK), as described by Lourenço et al. [48]. Added citrates, free water-soluble polyphosphates, and distinction of the types of added phosphates was determined in the fillets and in the thawed ice glaze water (ice glaze + drip loss) by high pressure ion exchange chromatography with conductivity detection (IEC) according to Dafflon, Scheurer, Gobet, & Bosset [49] and Nguyen et al. [50]. Since preparation of samples for IEC causes a deprotonation of citric acid to citrate, the IEC cannot distinguish citric acid and sodium citrate and, thus, the determined value represents the content of citrate. Analysis was made in a Dionex ICS-5000^+^ IEC system (Thermo Scientific, Sunnyvale, CA, USA) equipped with an online eluent generator RFIC-EG^®^ ECG III KOH and an Anion ERS 500 Suppressor. Separation was done with a 25–70 mm KOH gradient program at 0.25 mL/min for 60 m in a column IonPac AS16 (2 × 250 mm) at 30 °C and quantification performed by a Dionex conductivity detector. Calibration curves were prepared for PO_4_, P_2_O_7_, P_3_O_9_, P_3_O_10_, and citrates within the range 0.5 to 80 mg/L using H_2_NaO_4_P.H_2_O (Sigma), Na_4_O_7_P_2_.10H_2_O (Sigma-Aldrich), Na_3_O_9_P_3_ (Aldrich), Na_5_O_10_P_3_ (Sigma-Aldrich) and citric acid (Merck), respectively. Results are expressed as g P_2_O_5_/kg in the case of hake fillets and g P_2_O_5_/L in the case of thawing waters. In quality assurance of measurements, a maximum relative error of 5% was, in general, accepted for validation of methods precision and accuracy. 

#### 2.2.3. Hake Fillets Species Identification

Portions of muscle tissue of hake fillets were collected and preserved in 96% ethanol for molecular biology analyses. Total genomic DNA was extracted from the muscle tissue of fillets using magnetic bead separation technology from MPure Tissue DNA Extraction Kit (MP Biomedicals, Solon, OH, USA), following the manufacturer’s protocol. The 652 bp barcode region of the mitochondrial DNA gene cytochrome c oxidase (COI) was subsequently amplified using primer cocktail C_FishF1t1–C_FishR1t1 [25] under the following PCR protocol and cycling conditions: 4 min at 94 °C followed by 35 cycles of 94 °C (30 s), 52 °C (40 s), and 72 °C (1 min), and a final extension at 72 °C (10 min). Each PCR mixture included 12.5 µL of Thermo Scientific Maxima Hot Start Green PCR Master Mix (2X) containing Maxima^®^ Hot Start *Taq* DNA Polymerase, optimized Hot Start PCR buffer, 4 mm of Mg^2+^ and 400 mm of each dNTP, 0.125 µL of each forward and reverse primer (0.01 mm), 4 µL of DNA template and 8 µL of ultrapure water to fulfil 25 µL of total reaction volume. PCR products were checked on a 1.4% agarose gel and were then sequenced at STAB VIDA (Caparica, Portugal) using sequencing primers M13F and M13R [51]. Forward and reverse sequence chromatograms were assembled, edited and aligned using MEGA version 7.0.26 [52] and used for query in both NCBI BLAST (http://blast.ncbi.nlm.nih.gov/Blast.cgi, accessed on 20 November 2018) and Barcode of Life Data System—BOLD (http://v4.boldsystems.org/index.php/IDS_OpenIdEngine, accessed on 20 November 2018) to search for similarities in these databases. Identification was considered positive when 98% to 100% of sequence identity was observed. Identified species were compared with the ones referred in the packages to assess compliance.

#### 2.2.4. Microbiology Analysis

Microbiological testing involved enumeration of total aerobic microorganisms, sulphite-reducing bacteria, total coliforms and *E. coli*, yeast and mold, *Staphylococcus aureus*, and detection of *Salmonella* Sp. Samples for microbiological analysis of pre-packaged hake products were prepared aseptically according to ISO 7218:2007/AMD 1:2013 [53] and 6887-3:2017 [54]. For each sample, initial dilution was prepared by aseptically homogenizing 25 g of hake fillet with 225 mL maximum recovery diluent (MRD, Oxoid, UK) [55]. Adequate serial dilutions were prepared and bacterial counts were determined by the pour plate method except stated otherwise. The analysis were based in ISO standards or alternative published methods: enumeration of total aerobic microorganisms at 30 °C [56], enumeration of sulphite-reducing bacteria (internal method based on ISO 15213 (2003) by incorporation in Iron Agar (LYNGBY) without L-Cysteine (IA, Oxoid, UK), enumeration of total coliforms and *E. coli* (internal method based on ISO 4832:2006 and ISO 16649-2:2001 by incorporation in Chromocult Coliform Agar, CCA, Merck, Germany), surface plating for yeast and mold enumeration with Rose Bengal Chloramphenical Agar (RB, Sharlau, Spain) [57] and enumeration of *Staphylococcus aureus* with Baird Parker Agar supplemented with Tellurite Emulsion (BP, Oxoid, UK) [58]. Results were expressed as log colony-forming units per gram of sample (log cfu/g). For detection of *Salmonella* Sp. (Rapid Salmonella method, BioRad—AFNOR BRD 07/11–12/05) the initial enrichment solution was prepared with 25 g of fish flesh and 225 mL of Buffered Peptone Water (BPW, Oxoid, UK) supplemented with one capsule of Rapid’*Salmonella* (BioRad, Hercules, CA, USA) for selective enrichment and plated in Rapid’*Salmonella* Agar (RSA, Bio-Rad, Hercules, CA, USA). Suspected colonies were further isolated and confirmed using *Salmonella* latex agglutination test (Bio-Rad, USA) and results refer to presence/absence of *Salmonella* in 25 g of muscle. 

#### 2.2.5. Sensory Analysis

The packaging defects (presence of holes/abnormal traces), presence of ice and fillet’s size variation were evaluated by five assessors when the packages were opened using a five-point hedonic scale ranging from defect absent (1 point) to very intense (5 points). Sensory evaluation was done in a dedicated test room by a sensory panel composed of 10 experienced assessors on fish quality control, selected from the seafood sensory panel of the Portuguese Institute for the Sea and Atmosphere (IPMA, I.P.). Analysis of dehydration, muscle gaps, blood stains, odor, and color were performed with a five-point hedonic scale in a set of raw fillets, one from each package, mostly out of 10, and all of the same batch after being thoroughly thawed overnight in a refrigerator (4 °C ± 1 °C). Overall quality of raw products was determined as the average of the mean value of each evaluated sensory attribute together with the scores from packaging defects, fillet’s size variation, and presence of ice. For evaluation of cooked fillets, a piece (100 g ± 5 g) of the larger area from each thawed fillet was individually wrapped in aluminum foil (food use) and steam-cooked for 10 min at 100 °C in a steam oven (Combi-Master CM64; Rational, Lund, Sweden) with air circulation, without adding salt or spices. In each test session, three samples—still warm and wrapped for retention of aroma—were presented to the sensory panel and the assessors (10) evaluated the intensity of sensory attributes, color/appearance (typical, discoloration, dehydration), odor (typical, rancid), flavor (typical, rancid/bitter) and texture (firmness, succulence), using an affective method [59] involving a five-point hedonic scale, ranging from absent (1 point) to very intense (5 points). Assessors also evaluated the overall quality of the products (overall appreciation) using a five-point scale, ranging from bad (1 point) to excellent (5 points). The mean value of each evaluated attribute was determined and an average score of 3 (fair) considered the limit of acceptability.

### 2.3. Statistical Analysis

The results of the different quality parameters of hake fillet samples were evaluated together by factor analysis, using the principal components method (PCA), to identify groups of hake samples with similar quality characteristics and to identify the variables responsible for those differences. The parameters included in PCA were storage length, glaze ice, moisture, protein, M/P ratio, soluble protein, soluble/total protein, thaw drip loss, thaw drip loss protein, WHC_raw_, WHC_cook_, WHC_total_, TVB-N, pH, total phosphates in fillets, total phosphates in thawing waters, orthophosphates in fillets, orthophosphates in thawing waters, and sensory analysis (overall score of negative descriptors of raw hake fillets, overall score of positive descriptors of cooked hake fillets, and overall score of negative descriptors of cooked fillets). Polyphosphates and citrates contents were not considered for the multivariate analysis on account of the influence of zero data (values lower than LOQ or LOD) for the dispersion of samples points in the plot. Regarding the sensory attributes used in the PCA, in raw fillets, the overall score of negative descriptors was determined by calculating the mean values of the scores of color (yellow) and odor (rancid). In cooked fillets, the overall score of negative descriptors was determined by calculating the mean values of the scores of discoloration, dehydration, rancid odor, and rancid/bitter flavor, while the overall score of positive descriptors was determined by calculating the mean values of the scores of typical color, typical odor, typical flavor, and texture (firmness and succulence). The different quality parameters were also tested for correlation according to the Pearson correlation coefficient test. All statistical analyses were tested at a 0.05 level of probability. All statistical treatment was done with the software STATISTICA^©^ from StatSoft, Inc. (Tulsa, OK, USA) version 10, www.statsoft.com.

## 3. Results and Discussion

### 3.1. Hake Fillets Species Identification 

Replacement of more valuable fish species by cheaper ones is a current practice in today’s markets [60,61]. Given that there is only interest in evaluating products that can be sold under the commercial designation of hake, the first phase of the study involved the evaluation of the products, in relation to conformity of labeled fish species presented in Table 1. Most of the samples were in accordance with the label, either on genus or species (Table 2). Samples no. 5 and no. 20 were the exceptions and though the genus *Merluccius* was correctly identified in both samples, the identified species were *M. hubbsi* and *M. paradoxus*, respectively, instead of *M. capensis*. Regarding no. 20 mislabeling, it is known that in the same fishing ground *M. capensis* (species labeled) and *M. paradoxus* (identified species) coexist and that most reported catches combine both [62]. Because of difficulties in species separation during on-board processing, most processing companies list both species in the label to accommodate possible mixtures. Sample no. 11 (*Gadus chalcogrammus)* was also mislabeled as *M. hubbsi* and because not a hake species removed from the study. 

Overall, it is noteworthy the similarity of the level of conformity of the fish species labels in this study with the results obtained by Harris et al. [63], regarding specifically the hake products sold in Portuguese supermarkets. Besides the replacements detected in this study, hake has also been substituted by Panga (*Pangasius hypophthalmus*) as shown by Mendes and Silva [64]. Nevertheless, hake is not historically one of the species more likely to be replaced as it was observed in previous market studies [38,39]. 

### 3.2. Net Weight, Glaze Ice, Thaw-Drip, Cook Loss, and Water-Holding Capacity (Raw and Cook)

According to Portuguese regulation [37] deviations up <4% of the net weight are admissible and were therefore incorporated in the corrected net weight values (Table 3). Measurement of the net weight differences showed that 37% of the 19 commercial samples analyzed, presented mean negative deviations (1–15%) in relation to the weight indicated on the packaging. In terms of weight content, packages were in general uniform in the net weight (CV ≤ 4%) and only one sample (no. 18) showed a 7% weight variation between packages of the same brand. Nevertheless, the packages fillet’s size variation was relevant, with 79% of the packages showing between 50 and 75% of differently sized fillets (Table 3), which cannot be justified by the need to adjust a specific net weight at packaging.

No maximum or minimum added glaze ice has been regulated for fish and fishery products though addition of 5% to 10 % of the total weight should be sufficient to afford protection in most cases of properly wrapped and sealed products [65]. The values in hake samples ranged between 5.8% and 32.9% (Table 3) with a large majority of the products (79%) showing glazing levels higher than the recommended reference range. High amounts of ice glaze may be used to overcome the effects of larger storage periods or as an additional measure to compensate the low protection performance of the packaging; however, over-glaze can also be applied to hide defects, though it gives fillets an unpleasant appearance.

Determination of the thaw-drip loss and measurement of the changes in the water-holding capacity of fillets are some of the most straightforward analysis, used to evidence the decrease in the ability of the fish muscle to reabsorb the thawed water from melted ice crystals. Thaw-drip loss showed a wide range of variation in the hake fillets (1.2–25.1%), mostly due to the high levels observed in sample nos. 9, 15, and 16 (16.8–25.1%), all of the species *M. productus*. On the other hand, *M. hubbsi* (2.1–7.5%) and *M. paradoxus* (1.2–8.9%) samples showed relatively low levels and comparable dispersion of results. Similar range of variation of thaw-drip loss was determined by Ciarlo et al. [66] in *M. hubbsi* fillets stored at −20 °C for 10 months (1.7–6.7%). Miyauchi et al. [67] reported that the formation of thaw-drip in Pacific cod was dependent on storage temperature and time. Since values of 1–2% thaw-drip loss cannot be regarded as high [68], results showed that *M. productus* fillets were of significantly poorer quality than from other species, either because *M. productus* is characterized as one of the hake species with the softer texture [69] and/or due to incorrect manufacturing and storage procedures.

WHC_raw_ presented a 29% range of variation between samples (Table 3), with fillets of *M. paradoxus* in sample no. 1 displaying the highest WHC_raw_ level (68.2%) and *M. hubbsi* fillets of sample no. 13 showing the lowest (48.5 %). In mean terms fillets of *M. paradoxus* presented the highest WHC_raw_ level (55.9 ± 1.9%), followed by *M. productus* (52.2 ± 1.6%) and *M. hubbsi* (50.8 ± 1.9%). Though frozen storage length affects the WHC_raw_ [16,70] the levels of WHC_raw_ were uncorrelated with the fillet’s frozen storage time, possibly reflecting a multitude of other factors affecting this parameter, such as initial quality, muscle pH, temperature fluctuation, storage temperatures, or ice glaze [17,71].

While protein is denatured, the WHC of the muscle is reduced, therefore, when frozen fish is thawed, drip is usually produced and nutritive substances are carried away with the drip [66]. In comparison with thaw drip loss results the WHC_raw_ levels were relatively homogenous and showed a lower coefficient of variation, 9% versus 93%, possibly because water quantified in thaw drip loss is free water not connected in any way to membranes or cells walls and therefore, more easily released and more deeply reflecting changes in the muscle chemical and physical changes. On the contrary, WHC_raw_ is composed of expressible water extracted by a centrifugation force, overcoming the water-binding forces that limit free drip [72], and is therefore, more connected and less sensitive to reflect quality changes. Though it has been mentioned that high WHC results in low aggregation of protein and lower drip loss during thawing [73], no significant correlation was found in the fillet samples. 

Several variations of the WHC determination method in fishery products can be found (e.g., different times and centrifugation speeds), so results are not always comparable [74]. Using the same method, WHC_raw_ levels around 78% were reported to decrease to 53% in Argentine hake (*M. hubbsi*) fillets after 10 months at −20 °C [66]. A lower drop of WHC_raw_ levels in fillets of Atlantic hake (*M. merluccius*) from 60% to 43%, was observed after 5 months of frozen storage at −10 °C [75] and from 70% to 54% after 24 months at −20 °C [76]. On the other hand, a greater decrease was reported by Herrero et al. [70] from 71% to 48% also in Atlantic hake fillets at −10 °C in comparison to the observed from 71% to 58% at −30 °C, during a 9-month frozen storage period. Fillets of Mediterranean hake (*M. mediterraneus*) showed also an initial WHC level of 80% and a significant decrease to 57% after 12 months of frozen storage at −18 °C [77]. Evaluation of published data and consideration of the recommended practical storage life for lean fish, 9 months at −18 °C [9], point out to an acceptable level of WHC_raw_ around 58%, which was used as a quality standard in this work. Evaluation of samples WHC_raw_ levels, show that with the exception of samples no. 1 and no. 3 (*M. paradoxus*), all other samples failed this standard.

Hake fillets presented a relatively wide range of variation (13.3–22.9%) of cook loss. Samples showing the lowest cook loss levels (up to 15.1%) belonged all to the same species *M. hubbsi* (sample nos. 5, 10, 12, 13, and 17). Cooking losses have been reported to vary greatly with the fish species, the method of heating, and the heating regime [78]. Mean levels of cook loss were reported around 10% in *M. productus* [79] and 16.6% in *M. capensis* [80] analyzed with similar methodology. On the other hand, cook losses were accounted as 19% in *M. productus* after 1-month frozen storage at −29 °C [81]. Cook loss values in the same lower-level range as found in the hake fillet samples, were reported in several publications reviewed by Aitken and Connell [78]. 

Maximum water loss is reached when the muscle cell contracts due to denaturation of myosin and the extra-cellular spaces are widened [82]. Thus, hake fillet samples with higher cook loss levels than the reported in the literature, namely with cooking losses higher than 20%—as in sample nos. 4, 7, 19, and 20—are clearly indicative of poor-quality products. This may be the result of variable degrees of protein denaturation occurring during frozen storage for various reasons related with improper practices (e.g., temperature fluctuations, incorrect storage temperature, excessive frozen storage time).

With respect to the WHC in the cooked fillets (WHC_cook_) the levels ranged between 52.9% (sample no. 9) and 61.6% (samples no. 1 and no. 13). Variation of results between samples was considerably lower (15%) than the observed in WHC_raw_ possibly because the liquid lost during cooking (cook loss) was not measured in the determination. In general, WHC_cook_ was higher than WHC_raw_, possibly because of the effect of heat treatment on the resulting muscle protein denaturation, which leads to the appearance of some hydrophilic groups on the surface of protein molecules causing an increase in the water bound to such molecules [83]. At cooking temperatures, there is a relationship between the WHC of the fish muscle and the tissue specific heat-induced morphological changes, affected by the denaturation of myosin and other proteins [82]. Levels of WHC_cook_ around 72.5% were determined in fresh mullet (*Mugil cephalus*) [83], whereas 38% were reported in cod kept in ice for 2 days, after cooking at 70 °C [82]. Other reports refer WHC_cook_ between 70% and 80% in steamed fish [78]. In general, results determined in the frozen hake fillets were lower which can be indicative of products with reduced quality. 

### 3.3. pH

The measured pH values in hake fillets presented a range of variation between 6.58 and 7.15 (Table 4). Levels higher than 7.0 were determined in all samples of *M. productus* (nos. 9, 15, and 16), which showed also polyphosphates in the ice glaze. Though samples with higher pH levels were between the ones with higher storage time, no overall significant correlation between pH levels and storage time was found. Variation in the fillet’s initial quality and/or methods used for quality protection (e.g., ice glazing thickness, packaging quality, storage temperature) may mask this connection. An average pH level of 6.67 was reported in frozen *M. productus* [84] and an increase of pH levels shown to occur in Southwest Atlantic hake fillets during frozen storage from 6.8 to 7.4, after 10 months at −20 °C [66]. On the other hand, vacuum-packaged fillets samples of Mediterranean hake (*M. mediterraneus*) showed an initial pH of 6.45 and a significant increase to 6.94 after 12 months of frozen storage at −18 °C [77]. 

### 3.4. Soluble Protein, Soluble/Total Protein Percentage, and Thaw-Drip Protein

Soluble protein determined in the fillets showed a variation range between 9.7 and 13.2 g/100 g (Table 4). The highest value was detected in sample no. 1 (*M. paradoxus*) and the lowest values were detected in all samples of *M. productus*. Samples of this species displayed also the highest M/P ratio, 5.5 ± 0.2 (Table 4). Reported M/P ratio have been accounted as 4.7 ± 0.2 in *M. capensis/paradoxus*, 4.8 ± 0.2 in *M. hubbsi* and 5.0 ± 0.2 in *M. productus* [85]. Significant negative correlations (*p* < 0.01) were found between soluble protein and thaw-drip loss, and also between soluble protein and pH.

During frozen storage (9–10 months) of fillets at −20 °C, vacuum packaged *M. productus* showed a drop in the soluble protein levels from 18 g to 14 g/100 g [86], while in *M. hubbsi* packaged in cardboard boxes values drop from 19 g to 5 g/100 g [66], and in *M. capensis* packed in polyethylene bags decreased from 15 g to 5 g/100 g [87]. On the other hand, higher soluble protein levels were observed in whole frozen *M. productus* (22.2 g/100 g) and *M. merluccius* (17 g/100 g) [84,88]. Evaluation of the referenced data shows the existence of a high variability of reported soluble protein levels, particularly at the end of comparable storage times, which can be due to different experimental conditions (e.g., fish species, initial quality, type of packaging) and also, small differences in the methodologies used. Comparison of the present study data with published levels, shows high soluble protein contents, particularly in the fillets with the longest storage periods. This difference may be due to the contribution of non-protein nitrogen, not accounted in the referenced data because specific protein quantification methods were used on these (e.g., Lowry and Biuret methods). The NPN-fraction (non-protein nitrogen) in *M. productus* has been reported to be 16.8% of the total nitrogen in the muscle [84]. Furthermore, the short range of variation of soluble protein levels was also not related with the samples extended range of storage times (1–28 months), which was further evidenced by the nonexistence of a correlation between the two variables. The use of protective measures, like glaze ice or multilayer polyethylene bags may have reduced the variability in the soluble protein levels, but adulteration of real storage times is not set aside. 

In terms of the soluble/total protein percentage (SP/TP) a variation range between 59.7% and 72.2% (Table 4) was determined. Although it has been reported that the rate of deterioration when fish is held frozen varies markedly among species [16], no noticeable difference of the SP/TP percentage was detected between hake species, possibly because they are all of the same genus and/or there is an overlap of the variability arising from differentiated processing and storage procedures. Though this ratio is potentially more representative of changes occurring during frozen storage, since soluble protein levels are normalized to the total amount of protein in the fillets at the moment of analysis, no correlation was found with storage time or any other quality parameters. De Koning and Mol [87] reported a SP/TP percentage of 87.0% in shallow-water Cape hake (*M. capensis*) fillets at day 0 and a 27% after 10 months at −18 °C. On the other hand, levels around 94% were reported in fillets of *M. hubbsi* at day 0 and 63% and 41% after 3 and 6 months of frozen storage at −20 °C, respectively [66]. The SP/TP percentage was also reported to change in North Pacific hake (*M. productus*) fillets from 85% at the beginning of frozen storage to 22.3% after 120 days at −20 °C [89]. In comparison with data reported in the literature, the range of SP/TP percentage found in this study would correspond to a storage period of around 3 months at −18 °C. Nevertheless, the time of storage of the fillets was in general, significantly higher than 3 months and did not correlated with the SP/TP percentage, which by its turn did not correlate with any parameter, other than soluble protein. Likewise, protective packaging strategies could have an effect on the SP/TP percentage, as well as the use by the business operator in the fish fishery sector (processors and retailers) of lower than −18 °C storage conditions, but incorrection of labeled freezing dates is also a possibility.

In order to follow the denaturation and insolubilization of protein in the fillets, the protein in the water released by the fillets without ice glaze following a 24 h thawing, was also quantified (thaw-drip protein). Since most of the proteins found in drip are water-soluble, sarcoplasmic proteins [90], higher levels of thaw-drip protein were expected in samples with higher quality on account of higher solubilization of protein. Levels of thaw-drip protein presented a considerably wide range of variation, with values between 3.9 and 46.2 g/100 mL (Table 3). As in the case of soluble protein, the lowest levels (3.9–4.5 g/100 mL) were determined in all samples of North Pacific hake (*M. productus*), due to the dilution effect of the high levels of thaw-drip loss and thus, denoting inferior quality products. The other hake species presented a similar mean thaw-drip protein level around 17.7 g/100 mL. With the exception of a correlation with the thaw-drip loss (*p* < 0.05), no correlation was found with the fillet’s storage time or any other quality parameter. Reference thaw-drip protein results are very limited in the bibliography. A thaw-drip protein content of 8.5% and 7.7% was determined in Atka mackerel (*Pleurogrammus monopterygius*) fillets, thawed after 7- and 10-months storage (−18 °C), respectively [91].

### 3.5. Total Volatile Basic Nitrogen

TVB-N values in deep-frozen hake fillets ranged between 5.3 (no. 16) and 23.5 mg N/100 g (no. 13) (Table 4). Samples having lower than 10 mg N/100 g accounted for 15% of the total, whereas samples presenting a TVB-N content between ≥10 and 20 mg N/100 g were 79% of the total and >20 mg N/100 g were 5%. Based on former experience, hake products with TVB-N <10 mg/100 g are considered as having very good quality, between 10 mg and ≤20 mg/100 g of good quality, 20 mg to ≤35 mg/100 g inferior quality and >35 mg/100 g to be rejected. Despite the existence of some samples with relatively high TVB-N levels (nos. 4, 13, 14, 18, 19, 20), all samples showed TVB-N levels below the limit of 35 mg N/100 g, allowed for hake species [28]. An increase of the TVB-N levels of *M. mediterraneus* from 18 mg and 25 mg N/100 g to 33 mg and 42 mg N/100 g in whole fish and fillet, respectively after 12 months of frozen storage at −18 °C, was reported by Simeonidou et al. [77]. TVB-N levels were also found to increase from 21 mg to 28 mg N/100 g in *M. hubbsi* frozen stored during 10 months at −20 °C [66]. Considering that in freshly caught fish TVB-N content is generally >10 mg N/100 g and does not exceed 15 mg N/100 g [23], and also that TVB-N content increases, though at reduced rate during freezing temperatures because enzymes are still active, the levels determined in the hake fillets compare favorably with referenced data and are indicative of a reduced spoilage before freezing along with a generally low spoilage during frozen storage.

### 3.6. Total Phosphates and Free Polyphosphates 

In order to control the utilization of additives for moisture retention and acidity regulation, total phosphates, organic (PO_4_) and added inorganic (P_2_O_7_, P_3_O_9_ and P_3_O_10_) soluble phosphates were quantified (Table 5). Total phosphates (spectrophotometric method) ranged between 3.6 g and 5.2 g P_2_O_5_/kg in hake fillets. Though total phosphates analysis does not discriminate between organic phosphates and added inorganic phosphates, comparison with hake baseline levels reported by Teixeira et al. [92] in natural samples (4.8 g ± 0.5 g P_2_O_5_/kg), by Vlieg et al. [93] in hake (4.5 g P_2_O_5_/kg) from New Zealand waters and also reported levels in European hake (5.0 g P_2_O_5_/kg), South Africa hake (4.1 g P_2_O_5_/kg) and in Southern hake (4.1 g P_2_O_5_/kg) [94], allowed to conclude that the maximum permitted added level of 5 g P_2_O_5_/kg [35] was never reached in any sample. Furthermore, being the range of phosphate levels similar to the reference baseline levels, there is no evidence, in general, of their use in the processing of hake fillets, despite the labeling of E451 (triphosphate) as an ingredient in sample no. 16. 

Total phosphates were also quantified in the thawing water (glaze ice + drip loss) to find evidence of the use of phosphate-based additives in the processing of hake fillets (Table 5). Results ranged between 2.1 g and 5.1 g P_2_O_5_/L and showed a significant correlation (*p* < 0.01) with the total phosphates contents in the fillets. Samples no. 9, 15, and 16, all composed of *M. productus* presented the highest levels. Nevertheless, total phosphates determined in the glaze ice were, in most cases, possibly from natural origin and resulted from the drip loss and carry over of organic material during thawing of the fillets. Furthermore, since analysis of total phosphates using the spectrophotometric method does not distinguishes organic phosphates from added inorganic phosphates it was not possible to conclude unequivocally that phosphates were used. Not even in sample no. 16 labeled with E451 (triphosphate) as an ingredient. Quantification of free water-soluble polyphosphates and distinction of the types of added phosphates was done by ion exchange chromatography analysis with conductivity detection. This methodology detects the natural phosphates (PO_4_) and also added inorganic phosphates (P_2_O_7_; P_3_O_9_; P_3_O_10_), which are likely to degrade and convert into orthophosphate (PO_4_). Concentrations of orthophosphate (PO_4_) in the hake fillets ranged between 2.3 g and 4.6 g P_2_O_5_/kg showing samples of *M. productus* (nos. 9, 15, and 16) the highest levels. As in the case of total phosphates a significant correlation (*p* < 0.01) was found with the orthophosphate levels in the glaze ice, which showed a range of variation between 1.9 g and 3.8 g P_2_O_5_/L. In the case of diphosphate (P_2_O_7_), levels in the fillets were in general lower than the limit of detection (<0.02 g P_2_O_5_/kg) with exception of sample no. 15 that showed traceable amounts, though not quantifiable (LOQ = 0.05 g P_2_O_5_/kg). Levels lower than the limit of detection were obtained also in the glaze ice in all samples, with the exception of sample no. 15 again, and sample no. 9. In terms of triphosphates (P_3_O_9_ and P_3_O_10_) these were not detected in the fillets (<0.01 g P_2_O_5_/kg), but were evidenced in the glaze ice, particularly in sample nos. 9, 15, and 16, with levels of trimetaphosphate (P_3_O_9_) ranging between 1.0 g and 1.1 g P_2_O_5_/L and triphosphate (P_3_O_10_) up to 0.7 g P_2_O_5_/L. 

Phosphates based additives are composed of inorganic phosphates condensed synthetically that degrade into smaller phosphate molecules during processing. Though the origin of orthophosphate cannot be ascertained in relation to its nature (organic/inorganic), the detection and quantification of inorganic phosphates (diphosphate, trimetaphosphate, and triphosphate) particularly in the ice glaze of samples nos. 9, 15, and 16 is clear evidence that these phosphates were used in the processing of hake. Nevertheless, the maximum permitted added level phosphates was never reached in any sample.

From the point of view of labeling and evidence of the use of polyphosphates it is clear that despite the low levels of triphosphates determined in samples nos. 9, 15, and 16 these have been used, and should have been labeled in samples nos. 9 and 15 as was the case of sample no. 16. Detection of phosphates use in all samples of *M. productus* originated from the Northeast Pacific points to a single supplier of imported hake and to either (i) a possible traceability problem in the transmission of information regarding the use of phosphates; or (ii) a deliberate omission in the labels of phosphates as ingredient. 

### 3.7. Citrates

Evaluation of citrates addition to products is not easy to perform. On account of participation in physiological processes like the citric acid cycle, citric acid is naturally present in variable amounts in the muscle of fish and no limit allowing identification of addition to products has been defined. Furthermore, utilization of citrates in processed fishery products does not have a specific limit and is authorized by EU reg. 1333/2008 [35] as to be used *quantum satis*, with the requirement of correct labeling. All samples, both fillets and ice glaze, presented detectable citrate (citric acid and citrates) levels lower than the limit of quantification (LOQ = 0.024 g/kg), with exception of fillets in sample no. 6 that did not show detectable levels (<0.005 g/kg) (data not shown). Similar results (<LOQ) were previously obtained in controlled samples of *Merluccius merluccius* from the Portuguese coast not processed with citrates (data not published). 

Information in the literature about naturally occurring citric acid in aquatic animals is considerably limited. Bisenius et al. [95] reported citrate levels below the limit of detection (0.03 g/kg) in untreated cod fillets. In the case of cod fillets treated either with 2% citric acid (E330) or 2% sodium citrate (E331) levels of citrate around 16.3 g/kg were reached after 90 min. Manthey-Karl et al. [96] reported in wild and farmed Atlantic turbot, barramundi, and farmed pangasius citrate contents ranging from below the limit of detection (0.005 g/kg) to 0.03 g/kg. These authors further cite reports from the Swiss State Laboratory of the Canton Bern [97,98] mentioning that citric acid concentrations of citric acid lower than 0.1 g/kg determined in different processed frozen fish fillets and crustaceans must be of natural origin, although the Swiss scientists could not finally clarify, whether traces of citric acid were natural or from carry-over effects during production. Similarly, though being difficult to evaluate the use of citrates in the hake fillets of the present study, the extremely low levels detected point out to natural levels and absence of their use as additives in the manufacturing practices. In fact, with exception of sample no. 8 that labels citric acid (E331) as ingredient, none of the other samples has any indication regarding added citrates. Considering that none of the samples were possibly processed with citrates, labeling of these additives showed a 95% compliance since one sample listed E331 as ingredients and was not detected in quantities higher than the natural levels.

### 3.8. Microbiology Analysis

The overall microbiological quality of deep-frozen hake fillets was good (Table 6). Though there is no established limit for frozen hake, it is generally accepted a level of 5–6 log cfu/g for mesophilic bacteria in these products [30]. Total counts of aerobic mesophilic bacteria varied between 1–4 log cfu/g (Table 6). No *Escherichia coli* and *Staphylococcus aureus* were found in the samples. Less than 1 log cfu/g Total Coliforms were registered in 80% of the samples; for the remaining samples levels were <4 log cfu/g. *Salmonella* was not isolated in samples. H_2_S-producing bacteria constitute only a minor fraction of the initial microflora on newly caught fish [99] and have been used to evaluate the potential spoilage occurred during storage [100]. In a few samples, counts up to 2 log cfu/g of sulphite producers were found. Freezing does not destroy molds. Though there are no established limits for molds and yeasts for frozen fish, the presence of levels up to 2 log cfu/g of yeasts and molds in six samples indicate poor hygienic conditions in some of the production facilities during hake handling before freezing. In summary, microbiological quality of deep-frozen hake fillets sampled in the Portuguese retail market was generally good. Pathogens were absent in accordance with Commission Regulation (EC) no. 2073/2005 [29] and specific hygiene indicators were low and present in few samples within the guideline values established by ICMSF [30]. 

### 3.9. Packaging Analysis

Labeling was also analyzed regarding conformity with mandatory information requirements. In this regard, it is notable that all labels respect legal requisites (data not shown), even if they contain points to improve. Some opt for too small letters, others have a too elementary presentation. Labels do not always provide companies contact in Portugal; many do not contain nutritional and energy information and about a quarter of the samples do not indicate how thawing and/or preparation of product should be done. The producer of sample no. 17 does not discloses the brand. Like producer of sample no. 5, it does not mention the drained net weight, nor the fact that the product is not glazed (even if it is not mandatory because the packaging is transparent). Only one producer (sample no. 8) refers to the number of portions. Overall, there are still several aspects that producers need to improve in terms of information to consumers; namely, in what concerns all the recommendations and repairs mentioned.

Analysis of the labels in the hake packages (Table 1) shows a range of variation of the ‘best before’ period between 13 and 35 months and a mean recommended storage of 24 months. The majority of the companies (74%) use a best before date period higher than 20 months, being a period between 23–27 months the most frequently found. Even though some protection measures can increase the recommended storage (e.g., ice glaze, packaging material), it is clear that the majority of the hake fillet processors are using considerably higher ‘best before’ periods than recommended and thus, not following the best practices for frozen storage. Also, though laboratory analysis of the hake fillets was done before the end of the products ‘best before’ date, sensory evaluation showed a correlation with storage time (*p* < 0.05) and evidenced a majority of the hake samples well below standard quality (<3), thus pointing out to products that had suffered undesirable changes during frozen storage. In this regards, inadequate storage conditions at the retail shops may also be the responsible for these undesirable changes in initially good quality products. 

### 3.10. Sensory Analysis

Analysis of the packaging defects, despite the different quality of the wrapping materials used, showed no problems in this area with good to very good scores in all samples (Table 7), thus evidencing that all producers mastered the packaging technology. Presence of ice crystals inside the packages and dehydration signs on the fillets were relatively scarce in 95% of the samples (Table 7), only one sample showed extremely bad results, evidencing that in general packaging is efficient is this regard. Likewise, gaps and presence of blood stains in fillets were not a relevant issue, showing 95% and 89% of the samples, respectively, results between absent and moderate. Analysis of color and odor of raw fillets presented, however, more alarming results, with 27% and 42 % of the samples, respectively, presenting moderate to very intense yellowish discoloration and rancid odor. The overall quality of the majority (79%) of the raw fillets and packaging is good, but a significant number of products (21%) showed critical and disqualifying problems in the raw products, namely in relation to deterioration signs linked with oxidation evidenced by the fillet’s rancid odor and yellowish discoloration.

Sensory analysis of the cooked fillets showed that the majority of the products had very low ratings in the positive descriptors (Table 8), with scores ranging between absent to slight in typical color (69%), typical odor (90%), typical flavor (85%), firmness (52%), and succulence (74%). Also, in relation to the negative descriptors the sensory panel found in the cooked fillets various quality issues, namely signs of discoloration (yellowish/brownish), dehydration, rancid odor, and rancid/bitter flavor in 69%, 26%, 79%, and 48% of the samples, respectively. Results from the products overall quality analysis clearly evidences the significance of all these defects, with the assessors rating 63% of the products as having bad to poor quality and 37% of them with just fair quality. A significant positive correlation (*p* < 0.01) was found between soluble protein levels and the overall quality evaluation of cooked fillets, mostly reflecting the relevance of the correlation (*p* < 0.01) of succulence scores. Comparison of raw and cooked sensory evaluation showed considerably lower rating in cooked fillets. This difference is mainly explained by the contribution of taste scoring in cooked fillets and the effect on this of the volatilization of specific organic compounds induced by heat. Cooking enhances the smell and taste perception of these molecules and makes more detectable the degradation and oxidation flavors presented in the products. Since the fish fillets are consumed cooked, ultimately the attributes from this particular sensory experience and the characteristics that consciously or unconsciously the fish eater (test panel) considers should be present in a cooked product, that were taken into consideration for overall conclusion in detriment of the raw products evaluation. In summary, no product was rated as good or excellent, which is clear evidence that producers are still not succeeding in complying with best practices. Furthermore, it is significant that the majority of the producers are supplying products of bad and poor-quality and therefore, below the limit of acceptability defrauding consumers. 

### 3.11. Relationship between Quality Control Parameters Indices—Multivariate Analysis

A multivariate analysis was performed to all data obtained in this study in order to detect groups of commercial hake samples with similar quality characteristics and identify the most informative quality control parameters. The first three principal components (PCs) were plotted for the different commercial hake samples (Figure 1). The plot shows a separation of commercial hake samples related with the species identified. In general, samples of *M. paradoxus* showed lower PC1 values and higher PC2 values, while *M. productus* showed higher PC1 values, and *M. hubbsi* showed lower PC1 and PC2 values. This separation was due to the fact that samples of *M. paradoxus* and *M. hubbsi* showed higher values of protein content and overall score of attributes of cooked fillets, when compared with *M. productus*. Also, samples of *M. paradoxus* showed higher values of soluble proteins, soluble/total protein, and WHC_raw_. In its turn, the separation of *M. hubbsi* samples was related with the higher values of TVB-N. On the other hand, *M. productus* samples had higher values of thaw-drip loss, moisture, M/P ratio, total phosphates in fillets and thawing waters, and pH. Considering also the PC3 scores of samples, hake fillets with lower values of PC3 was related with higher values of WHC_cook_. 

The differentiation between species detected in this work is further supported by intrinsic quality distinction among hake species, affecting for example texture, which has been reported to vary from soft to moderately firm [69]. In this regard, *M. capensis*/*M. paradoxus* are recognized as having the firmest texture of the hake species, followed by *M. merluccius* and *M. hubbsi* as the top firmer species, whereas *M. productus* on the contrary is characterized by its very soft flesh [69,101].

The first PC was correlated with most variables (Table 9), being the strongest correlations found with thaw drip loss, moisture, M/P ratio, pH, total phosphates in fillets and thawing water, orthophosphates in fillets, and overall score of defects of cooked hake fillets (loadings were higher than 0.70) and also with protein, soluble protein, and overall score of attributes of cooked hake fillets (loadings were lower than −0.70). The strongest correlations with the second PC were observed for glaze ice and soluble/total protein (loadings were −0.70 and 0.75, respectively). In its turn, the third PC was correlated with WHC total and WHC_cook_ (loadings were −0.79 and −0.66, respectively).

The proximity of the projection of the variables in the plot of PC (results not shown) suggests a strong correlation between several variables, which was then confirmed by Pearson’s correlation coefficient. Strong correlations were found between thaw drip and M/P ratio (r = 0.81). Additionally, total phosphates in thawing waters and total phosphates in hake fillets were strongly correlated (r = 0.85), as well as total phosphates in thawing waters and moisture content of hake fillets (r = 0.83). Overall score defects of cooked fillets correlated with overall score defects of raw fillets (r = 0.87) and with overall score attributes of cooked fillets (r = −0.90).

The significant correlations determined between objective and subjective quality control parameters are presented in Table 10. Amerine et al. [102] and Kramer and Twigg [103] recommend that for quality control applications the correlation coefficient between objective and subjective determinations should be at least 0.80 (*p* < 0.01), though those greater than these are preferable. In this regard, overall sensory quality of cooked fillets correlated at the highest level of significance (*p* < 0.01) with M/P ratio, protein, and soluble proteins (Table 8). Considering an overall sensory quality score ≥ 3 (mean quality) as an acceptability limit, the corresponding value of these objective parameters are 4.3%, 18.2%, and 12.2%, respectively, which can be considered for the analyzed hake species as indicative limit levels of mean quality products. M/P ratio and protein are related with the intrinsic quality of the species and soluble protein with the quality changes during processing and frozen storage. Other correlations were determined and corresponding acceptable limits estimated (Table 10), however though the determined levels find support in referenced works (e.g., pH: 6.7; thaw drip loss: 3.4%), these are merely informative and need an increased size of the sample for confirmation, due to the inferior level of significance of the correlations (*p* < 0.05).

## 4. Conclusions

The evaluation of the overall quality of frozen hake fillets showed that, although the production and marketing of good quality frozen hake products can be warranted by the existence of adequate technological solutions, the practices by a significant part of the business operators in the fishery sector (processors and/or retailers) are still deficient and lacking behind the best standards.

Though the compliance of products in terms of mandatory information in the labels, mislabeling affected most of the samples, mostly because of declared weights higher than the observed, exceeding the regulated tolerances, incorrect hake species identification in the packages and non-indication of polyphosphates as an ingredient. Furthermore, products had in general a higher than recommended amount of ice glaze, low levels of soluble protein and undue thaw-drip loss, the latest both indicative of protein denaturation/aggregation and low-quality products. Also, some products had been stored frozen for excessively long periods and most of the indicated ‘best before’ periods are longer than recommended at −18 °C. In relation to the microbiological contamination pathogens were absent and though some hygiene indicators were low, the presence of coliform, *E. coli,* molds and yeasts in some samples indicate poor hygienic conditions in a number of production facilities during hake fillets handling before freezing. Nevertheless, the global microbiological quality of hake fillets was good and products do not present any significant health safety issues.

Regarding the sensory quality of the products critical for consumer’s acceptance and consumption, no product was rated as good or excellent. Correlation of this subjective quality parameter was found to have a higher level of significance with the M/P ratio, total and soluble protein.

In view of the extension of the non-conformities identified, it is clear that some hake fillet business operators need to proceed with an improvement of production standards and/or storage conditions. Likewise, though labeling respect European regulations it has still several aspects that producers need also to improve in terms of information to consumers (e.g., readability, nutritional and energy information, how to thaw and/or prepare the product). Considering also that the deficient processing detected in this study may be widespread through other countries consumers of frozen hake fillets and white fish, a larger market study on the quality of these products, should be performed to determine and confirm the full extension of the utilization of these deceptive practices in the European Market. Additionally, control of the quality of these products by official authorities should be further enforced in order to protect consumers and fair trade.

## Figures and Tables

**Figure 1 foods-10-00848-f001:**
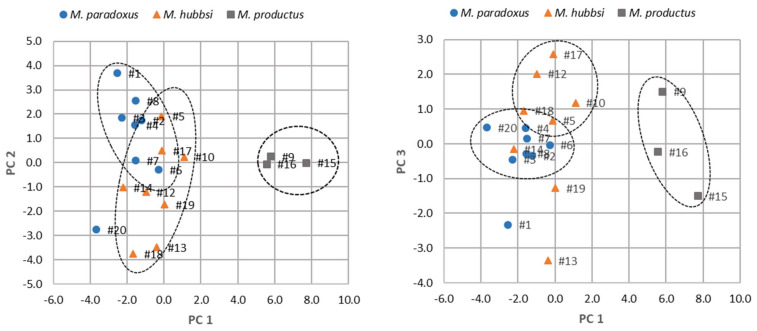
Principal components analysis (PCA) of quality control parameters of commercial hake fillets samples. PC 1, PC 2, and PC 3 explained 44.5%, 19.2%, and 9.8%, respectively, of the variation of the original variables. Numbers next to ‘#’ correspond to the sample number.

**Table 1 foods-10-00848-t001:** Deep-frozen pre-packaged hake products sampled in the Portuguese retail market.

Sample	Labeled Net Weight (kg)	Labeled Species	Capture Area	Freezing Date	Best Before	Storage Days *	Labeled Ingredients
1	400	*Merluccius capensis/paradoxus*	Southeast Atlantic	16 November 2017	10.2019	315	-
2	400	*Merluccius capensis/paradoxus*	Southeast Atlantic	06 September 2017	12.2019	385	Fillets of Cape hake
3	400	*Merluccius capensis/paradoxus*	Southeast Atlantic	10 July 2018	01.2020	81	-
4	400	*Merluccius capensis/paradoxus*	Southeast Atlantic (FAO 47)	23 July 2018	01.2020	68	Shallow-water Cape hake/Deep-water Cape hake
5	400	*Merluccius capensis*	Southeast Atlantic	02 May 2018	05.2020	149	Fillet, water
6	400	*Merluccius capensis/paradoxus*	Southeast Atlantic (FAO 47)	01 February 2018	02.2020	240	Cape hake
7	400	*Merluccius capensis/paradoxus*	Southeast Atlantic	08 August 2018	02.2020	53	-
8	600	*Merluccius capensis/paradoxus*	Southeast Atlantic (FAO 47)	28 November 2017	01.2020	303	Cape hake
9	600	*Merluccius productus*	Northeast Pacific	05 November 2017	11.2019	326	-
10	600	*Merluccius hubbsi*	Southwest Atlantic (FAO 41)	01 June 2016	05.2019	840	Argentine hake, fish
11	800	*Merluccius hubbsi*	Southwest Atlantic	20 July 2017	07.2019	431	-
12	662	*Merluccius hubbsi*	Southwest Atlantic	16 December 2018 **	02.2020	285	Hake, water
13	400	*Merluccius hubbsi*	Southwest Atlantic	21 September 2017	09.2019	370	-
14	800	*Merluccius hubbsi*	Southwest Atlantic	03 September 2018	09.2020	28	-
15	800	*Merluccius productus*	Northeast Pacific (FAO 67)	12 October 2016	10.2018	709	Hake, water
16	800	*Merluccius productus*	Northeast Pacific (FAO 67)	18 August 2017	05.2020	403	Hake, E451, water
17	1000	*Merluccius hubbsi*	Southwest Atlantic (FAO 41)	01 July 2017	01.2020	450	Fish
18	750	*Merluccius hubbsi*	Southwest Atlantic (FAO 41)	02 April 2018	05.2019	179	Fish, E330 and sugars
19	900	*Merluccius hubbsi*	Southwest Atlantic	06 August 2017	08.2019	415	Fish
20	700	*Merluccius capensis*	Southeast Atlantic	05 February 2018	07.2019	236	Fillets, water

* Until analysis. ** Labeling error, showing freezing date after sampling date.

**Table 2 foods-10-00848-t002:** Species identification of deep-frozen pre-packaged hake products sampled in the Portuguese retail market (*n* = 3).

Sample	Labeled Species	Sequence (GenBank)	Identified Species (*n* = 3)	Conformity (C/NC) *
1	*Merluccius capensis/paradoxus*	KP975790	*Merluccius paradoxus*	*C*
2	*Merluccius capensis/paradoxus*	KP975790	*Merluccius paradoxus*	*C*
3	*Merluccius capensis/paradoxus*	KP975790	*Merluccius paradoxus*	*C*
4	*Merluccius capensis/paradoxus*	KP975790	*Merluccius paradoxus*	*C*
5	*Merluccius capensis*	EU074469	*Merluccius hubbsi*	*NC*
6	*Merluccius capensis/paradoxus*	KP975790	*Merluccius paradoxus*	*C*
7	*Merluccius capensis/paradoxus*	KP975790	*Merluccius paradoxus*	*C*
8	*Merluccius capensis/paradoxus*	KP975790	*Merluccius paradoxus*	*C*
9	*Merluccius productus*	FJ164843	*Merluccius productus*	*C*
10	*Merluccius hubbsi*	EU074469	*Merluccius hubbsi*	*C*
11	*Merluccius hubbsi*	KX119441	*Gadus chalcogrammus*	*NC*
12	*Merluccius hubbsi*	EU074469	*Merluccius hubbsi*	*C*
13	*Merluccius hubbsi*	EU074469	*Merluccius hubbsi*	*C*
14	*Merluccius hubbsi*	EU074469	*Merluccius hubbsi*	*C*
15	*Merluccius productus*	FJ164843	*Merluccius productus*	*C*
16	*Merluccius productus*	FJ164843	*Merluccius productus*	*C*
17	*Merluccius hubbsi*	EU074469	*Merluccius hubbsi*	*C*
18	*Merluccius hubbsi*	EU074469	*Merluccius hubbsi*	*C*
19	*Merluccius hubbsi*	EU074469	*Merluccius hubbsi*	*C*
20	*Merluccius capensis*	KP975790	*Merluccius paradoxus*	*NC*

* C—Conform; NC—Non-Conform.

**Table 3 foods-10-00848-t003:** Corrected net weight, net weight difference, glaze ice (*n* = 10), thaw drip and cook loss and water holding capacity (raw/cook) levels (*n* = 3) in deep frozen pre-packaged hake products sampled in the Portuguese retail market. Results expressed as mean and standard deviation.

Sample	Corrected Net Weight (g) *	Net Weight Difference (%)	Glaze Ice (%) *	Thaw Drip Loss (%)	WHC–Raw (%)	Cook Loss (%)	WHC–Cook (%)
1	402.1 ± 5.7	0.5 ± 1.4	7.6 ± 0.9	4.9 ± 3.1	68.2 ± 2.4	15.3 ± 0.7	61.6 ± 0.3
2	409.5 ± 13.1	2.4 ± 3.3	14.3 ± 1.8	1.2 ± 1.1	56.0 ± 0.6	17.3 ± 3.1	58.2 ± 0.1
3	397.5 ± 6.1	−0.6 ± 1.5	13.2 ± 1.0	4.6 ± 1.0	60.1 ± 1.1	17.3 ± 1.3	60.0 ± 0.9
4	419.2 ± 6.8	4.8 ± 1.7	12.5 ± 1.3	3.1 ± 1.9	54.0 ± 1.5	21.5 ± 2.6	58.0 ± 1.7
5	418.0 ± 15.2	4.5 ± 3.8	10.0 ± 0.7	2.1 ± 2.0	53.0 ± 2.2	15.1 ± 3.1	55.6 ± 0.2
6	413.3 ± 15.0	3.3 ± 3.7	13.4 ± 1.5	5.2 ± 1.6	50.5 ± 2.2	17.3 ± 3.5	56.9 ± 1.6
7	411.4 ± 11.5	2.9 ± 2.9	14.9 ± 1.8	8.9 ± 2.3	54.0 ± 0.4	20.8 ± 3.1	58.8 ± 0.1
8	643.8 ± 7.3	7.3 ± 1.2	5.8 ± 0.6	4.7 ± 2.5	56.2 ± 2.2	16.9 ± 1.4	59.9 ± 0.2
9	601.1 ± 11.0	0.2 ± 1.8	19.3 ± 1.5	25.1 ± 0.9	51.5 ± 1.0	17.2 ± 2.2	52.4 ± 2.1
10	629.6 ± 25.4	4.9 ± 4.2	7.7 ± 1.6	7.5 ± 4.0	49.0 ± 0.7	14.9 ± 1.2	55.3 ± 0.7
12	559.8 ± 14.6	−15.4 ± 2.2	25.2 ± 1.9	5.4 ± 3.7	54.0 ± 1.6	13.3 ± 4.1	56.5 ± 0.0
13	381.7 ± 13.3	−4.6 ± 3.3	18.3 ± 3.0	2.4 ± 1.0	48.5 ± 1.2	14.8 ± 4.1	61.6 ± 1.1
14	793.6 ± 29.2	−0.8 ± 3.6	17.0 ± 2.0	2.4 ± 3.2	50.3 ± 1.9	19.6 ± 1.3	60.2 ± 0.9
15	723.5 ± 26.6	−9.6 ± 3.3	23.0 ± 3.2	16.8 ± 4.2	51.0 ± 0.6	19.3 ± 3.9	55.5 ± 1.0
16	756.9 ± 20.5	−5.4 ± 2.6	28.3 ± 1.8	22.6 ± 2.2	54.0 ± 1.5	15.8 ± 2.0	58.0 ± 0.4
17	1034.3 ± 32.8	3.4 ± 3.3	12.1 ± 0.6	3.0 ± 1.3	50.5 ± 1.1	15.0 ± 1.0	54.4 ± 0.0
18	714.1 ± 49.3	−4.8 ± 6.6	32.9 ± 3.8	4.9 ± 5.1	50.5 ± 1.2	17.6 ± 0.6	59.6 ± 0.7
19	916.3 ± 20.6	1.8 ± 2.3	12.1 ± 1.9	6.3 ± 2.4	49.1 ± 0.4	20.2 ± 2.1	59.2 ± 1.2
20	746.6 ± 26.2	6.7 ± 3.7	27.8 ± 2.2	7.5 ± 3.2	54.5 ± 1.0	22.9 ± 2.3	60.0 ± 0.6

* According to Portuguese Regulation Decreto-Lei no. 37/2004 [37].

**Table 4 foods-10-00848-t004:** Moisture and protein content, M/P ratio, TVB-N, soluble protein, soluble/total protein percentage, and thaw drip loss protein levels in deep frozen pre-packaged hake products sampled in the Portuguese retail market. Results expressed as mean and standard deviation (*n* = 3).

Sample	Moisture (%)	Protein (%)	M/P Ratio	pH	TVB-N (mg/100 g)	Soluble Protein (g/100 g)	Soluble/Total Protein (%)	Thaw Drip Loss Protein (g/100 mL)
1	79.7 ± 0.0	19.5 ± 0.7	4.1	6.73 ± 0.03	9.3 ± 1.1	13.2 ± 0.0	67.5 ± 0.1	12.7 ± 0.2
2	80.7 ± 0.3	17.3 ± 0.7	4.7	6.73 ± 0.02	11.7 ± 1.4	12.4 ± 0.3	71.5 ± 1.5	46.2 ± 0.5
3	79.1 ± 0.0	18.2 ± 0.3	4.4	6.58 ± 0.00	11.6 ± 1.4	12.4 ± 0.0	68.5 ± 0.2	16.9 ± 0.3
4	80.0 ± 0.6	17.2 ± 0.4	4.6	6.74 ± 0.04	16.2 ± 1.9	12.4 ± 0.1	72.2 ± 0.5	21.7 ± 0.1
5	80.8 ± 0.4	17.7 ± 0.7	4.6	6.84 ± 0.05	12.5 ± 1.5	12.0 ± 0.0	67.8 ± 0.2	21.9 ± 0.2
6	80.6 ± 0.0	17.9 ± 0.2	4.5	6.70 ± 0.03	14.3 ± 1.7	11.8 ± 0.1	65.9 ± 0.6	13.0 ± 0.3
7	78.9 ± 0.4	18.2 ± 0.4	4.3	6.57 ± 0.00	10.7 ± 1.3	11.8 ± 0.0	64.7 ± 0.2	10.2 ± 0.2
8	80.7 ± 0.0	17.8 ± 0.2	4.5	6.74 ± 0.04	11.3 ± 1.4	12.6 ± 0.3	70.7 ± 1.6	9.1 ± 0.3
9	83.0 ± 0.3	15.5 ± 0.3	5.4	7.02 ± 0.02	11.6 ± 1.4	9.8 ± 0.0	63.4 ± 0.1	3.9 ± 0.1
10	79.4 ± 0.9	17.0 ± 0.2	4.7	6.77 ± 0.02	11.9 ± 1.4	11.7 ± 0.2	68.8 ± 0.9	7.2 ± 0.0
12	79.6 ± 0.4	17.4 ± 0.3	4.6	6.85 ± 0.00	14.9 ± 1.8	11.6 ± 0.3	66.3 ± 1.6	14.5 ± 0.0
13	81.3 ± 0.1	17.9 ± 0.4	4.6	6.94 ± 0.00	23.5 ± 2.8	11.0 ± 0.1	61.4 ± 0.8	31.7 ± 0.2
14	78.7 ± 0.1	18.1 ± 0.0	4.4	6.85 ± 0.01	17.0 ± 2.0	11.7 ± 0.3	64.7 ± 1.5	24.6 ± 0.3
15	82.5 ± 0.4	15.5 ± 0.1	5.3	7.00 ± 0.01	7.1 ± 0.9	9.7 ± 0.3	62.9 ± 2.2	4.5 ± 0.1
16	84.1 ± 0.5	14.6 ± 0.2	5.8	7.15 ± 0.03	5.3 ± 0.6	9.9 ± 0.3	67.7 ± 1.9	4.6 ± 0.0
17	78.6 ± 0.1	17.2 ± 0.3	4.6	6.86 ± 0.07	11.9 ± 1.4	12.0 ±0.5	69.5 ± 2.7	14.2 ± 0.3
18	78.7 ± 1.4	17.1 ± 0.0	4.6	6.95 ± 0.05	15.4 ± 1.8	10.9 ±0.4	63.7 ± 2.5	15.1 ± 0.2
19	78.6 ± 0.7	18.0 ± 0.6	4.4	6.88 ± 0.01	17.0 ± 2.0	11.2 ± 0.2	62.6 ± 1.2	9.7 ± 0.2
20	78.4 ± 0.4	18.6 ± 0.2	4.2	6.58 ± 0.03	18.8 ± 2.3	11.1 ± 0.3	59.7 ± 1.8	13.8 ± 0.2

**Table 5 foods-10-00848-t005:** Total phosphates, free orthophosphate (PO_4_), pyrophosphate (P_2_O_7_), triphosphates (P_3_O_9_ and P_3_O_10_) in deep frozen pre-packaged hake fillets and thaw water sampled in the Portuguese retail market. Results expressed as mean and standard deviation (*n* = 3). Total phosphates were measured with a spectrophotometric method and free phosphates with an ion exchange chromatographic method.

	Total Phosphates		PO_4_		P_2_O_7_ *		P_3_O_9_ *		P_3_O_10_ *	
Sample	Fillets (g P_2_O_5_/kg)	Thaw Water (g P_2_O_5_/L)	Fillets (g P_2_O_5_/kg)	Thaw Water (g P_2_O_5_/L)	Fillets (g P_2_O_5_/kg)	Thaw Water (g P_2_O_5_/L)	Fillets (g P_2_O_5_/kg)	Thaw Water (g P_2_O_5_/L)	Fillets (g P_2_O_5_/kg)	Thaw Water (g P_2_O_5_/L)
1	4.3 ± 0.2	3.9 ± 0.1	3.5 ± 0.1	3.5 ± 0.0	<LOD	<LOD	<LOD	<LOD	<LOD	<LOD
2	4.3 ± 0.1	3.5 ± 0.1	3.3 ± 0.1	3.0 ± 0.1	<LOD	<LOD	<LOD	<LOD	<LOD	<LOD
3	4.2 ± 0.0	3.4 ± 0.0	3.2 ± 0.0	3.1 ± 0.0	<LOD	<LOD	<LOD	<LOD	<LOD	<LOD
4	4.2 ± 0.1	3.5 ± 0.1	3.2 ± 0.0	3.1 ± 0.1	<LOD	<LOD	<LOD	<LOD	<LOD	<LOD
5	4.7 ± 0.1	3.7 ± 0.1	3.4 ± 0.1	3.4 ± 0.0	<LOD	<LOD	<LOD	<LOD	<LOD	<LOD
6	4.2 ± 0.0	3.3 ± 0.0	3.1 ± 0.0	2.9 ± 0.0	<LOD	<LOD	<LOD	<LOD	<LOD	<LOD
7	4.2 ± 0.0	3.2 ± 0.0	3.2 ± 0.0	2.7 ± 0.2	<LOD	<LOD	<LOD	<LOD	<LOD	<LOD
8	4.3 ± 0.0	3.4 ± 0.0	3.5 ± 0.0	3.1 ± 0.1	<LOD	<LOD	<LOD	<LOD	<LOD	<LOD
9	4.8 ± 0.0	5.1 ± 0.0	3.9 ± 0.3	3.3 ± 0.3	<LOD	<LOQ	<LOD	1.1 ± 0.0	<LOD	0.3 ± 0.1
10	4.3 ± 0.0	3.4 ± 0.0	3.1 ± 0.2	2.9 ± 0.2	<LOD	<LOD	<LOD	<LOD	<LOD	<LOD
12	4.0 ± 0.1	2.5 ± 0.0	3.0 ± 0.0	2.3 ± 0.1	<LOD	<LOD	<LOD	<LOD	<LOD	<LOD
13	3.8 ± 0.1	3.4 ± 0.0	2.7 ± 0.1	2.9 ± 0.1	<LOD	<LOD	<LOD	<LOD	<LOD	<LOD
14	4.1 ± 0.0	3.0 ± 0.0	3.0 ± 0.2	2.7 ± 0.0	<LOD	<LOD	<LOD	<LOD	<LOD	<LOD
15	5.2 ± 0.0	4.9 ± 0.0	4.6 ± 0.1	3.8 ± 0.0	< LOQ	<LOQ	<LOD	1.0 ± 0.1	<LOD	0.7 ± 0.0
16	4.4 ± 0.0	4.6 ± 0.0	3.9 ± 0.0	3.0 ± 0.0	<LOD	<LOD	<LOD	1.1 ± 0.1	<LOD	<0.04
17	4.3 ± 0.1	3.0 ± 0.2	3.2 ± 0.0	2.6 ± 0.2	<LOD	<LOD	<LOD	<LOD	<LOD	<LOD
18	3.6 ± 0.0	2.1 ± 0.0	2.6 ± 0.1	1.9 ± 0.1	<LOD	<LOD	<LOD	<LOD	<LOD	<LOD
19	4.2 ± 0.1	3.4 ± 0.0	2.9 ± 0.1	3.0 ± 0.0	<LOD	<LOD	<LOD	<LOD	<LOD	<LOD
20	3.6 ± 0.0	2.5 ± 0.0	2.3 ± 0.0	2.2 ± 0.0	<LOD	<LOD	<LOD	<LOD	<LOD	<LOD

* P_2_O_7_: LOQ = 0.054 g P_2_O_5_/kg; LOD = 0.016 g P_2_O_5_/kg; P_3_O_9_: LOQ = 0.014 g P_2_O_5_/kg; LOD = 0.004 g P_2_O_5_/kg; P_3_O_10_: LOQ = 0.043 g P_2_O_5_/kg; LOD = 0.013 g P_2_O_5_/kg. LOQ—limit of quantification; LOD—limit of detection.

**Table 6 foods-10-00848-t006:** Counts (cfu/g) and presence (*Salmonella*) of microorganisms in frozen pre-packaged hake fillets sampled in the Portuguese retail market (*n* = 3).

Sample	Aerobic Mesophilic 30 °C (cfu/g)	Sulphite Producers (cfu/g)	Total Coliforms (cfu/g)	*Escherichia coli* (cfu/g)	Yeast and Molds (cfu/g)	*Staphylococcus Aureus* (cfu/g)	Salmonella (25 g)
1	1.2 × 10^3^	<1.0 ×10	<1.0 × 10	<1.0 × 10	5.3 × 10	<1.0 × 10	Negative
2	6.0 × 10	<1.0 × 10	<1.0 × 10	<1.0 × 10	<1.0 × 10	<1.0 × 10	Negative
3	5.8 × 10^3^	<1.0 × 10	<1.0 × 10	<1.0 × 10	<1.0 × 10	<1.0 × 10	Negative
4	8.6 × 10^3^	2.5 × 10^2^	<4.0 × 10	<1.0 × 10	<4.0 × 10	<1.0 × 10	Negative
5	6.5 × 10^2^	<1.0 × 10	<1.0 × 10	<1.0 × 10	<4.0 × 10	<1.0 × 10	Negative
6	1.2 × 10^3^	<4.0 × 10	<1.0 × 10	<1.0 × 10	<4.0 × 10	<1.0 × 10	Negative
7	3.0 × 10^3^	1.1 × 10	<1.0 × 10	<1.0 × 10	<1.0 × 10	<1.0 × 10	Negative
8	6.2 × 10^2^	1.5 × 10^2^	<1.0 × 10	<1.0 × 10	<4.0 × 10	<1.0 × 10	Negative
9	4.0 × 10	<1.0 ×10	<1.0 × 10	<1.0 × 10	1.6 × 10	<1.0 × 10	Negative
10	3.6 × 10	<4.0 × 10	<1.0 × 10	<1.0 × 10	<4.0 × 10	<1.0 × 10	Negative
12	2.8 × 10^3^	<4.0 × 10	<1.0 × 10	<1.0 × 10	4.4 × 10^2^	<1.0 × 10	Negative
13	6.1 × 10^3^	<4.0 × 10	<1.0 × 10	<1.0 × 10	<1.0 × 10	<1.0 × 10	Negative
14	6.2 × 10^2^	<1.0 × 10	<1.0 × 10	<1.0 × 10	3.6 × 10	<1.0 × 10	Negative
15	1.6 × 10^2^	<1.0 × 10	<1.0 × 10	<1.0 × 10	<4.0 × 10	<1.0 × 10	Negative
16	4.7 × 10^2^	<1.0 × 10	<1.0 × 10	<1.0 × 10	<1.0 × 10	<1.0 × 10	Negative
17	1.4 × 10^2^	<1.0 × 10	<1.0 × 10	<1.0 × 10	<4.0 × 10	<1.0 × 10	Negative
18	6.5 × 10^3^	<4.0 × 10	<1.0 × 10	<1.0 × 10	<4.0 × 10	<1.0 × 10	Negative
19	5.0 × 10^4^	<4.0 × 10	<1.0 × 10	<1.0 × 10	1.7 × 10^2^	<1.0 × 10	Negative
20	3.5 × 10^3^	<4.0 × 10	<4.0 × 10	<1.0 × 10	5.7 × 10	<1.0 × 10	Negative

**Table 7 foods-10-00848-t007:** Sensory analysis of raw hake fillets. Frequency distribution (%) of attributes and overall quality scores of 19 commercial hake fillet products (*n* = 10).

SENSORY SCALE					
ATTRIBUTES	Absent (1)	Slight (2)	Moderate (3)	Intense (4)	Very Intense (5)
Packaging defects	89	11			
Size variation *		21	74	5	
Presence of ice	69	26			5
Dehydration	69	26			5
Muscle gaps	42	37	16		5
Blood stains	5	53	31	11	
Color (yellow)	15	58	11	11	5
Odor (rancid)	21	37	11	26	5
**OVERALL QUALITY**	**Excellent**	**Good**	**Fair**	**Poor**	**Bad**
	5	74	16	5	

* Five-point scale: 1—all fillets of the same size; 2—25% of fillets with different size; 3—50% of fillets with different size; 4—75% of fillets with different size; 5—all fillets of different size.

**Table 8 foods-10-00848-t008:** Sensory analysis of cooked hake fillets. Frequency distribution (%) of negative attributes (discoloration, dehydration, rancid odor, and rancid/bitter flavor), positive attributes (typical color/appearance, typical odor, typical flavor, firmness, and succulence), and overall quality scores of 19 commercial hake fillet products.

SENSORY SCALE					
NEGATIVE ATTRIBUTES	Very Intense	Intense	Moderate	Slight	Absent
Discoloration			16	53	31
Dehydration			5	21	74
Rancid odor		5	11	63	21
Rancid/bitter flavor			16	32	53
**POSITIVE ATTRIBUTES**	**Absent**	**Slight**	**Moderate**	**Intense**	**Very Intense**
Typical color/appearance		69	31		
Typical odor	21	69	10		
Typical flavor	16	69	15		
Firmness	5	47	48		
Succulence	10	64	26		
**OVERALL QUALITY**	**Bad**	**Poor**	**Fair**	**Good**	**Excellent**
	16	47	37		

**Table 9 foods-10-00848-t009:** Loadings of principal component analysis of quality control parameters of commercial hake fillet samples.

Parameters	PC 1	PC 2	PC 3
Storage length	0.57	−0.02	−0.05
Glaze ice	0.24	**−0.70**	0.14
Moisture	**0.80**	0.17	−0.21
Protein	**−0.88**	0.10	−0.25
M/P ratio	**0.91**	−0.03	0.12
Soluble protein	**−0.78**	0.58	−0.03
Soluble/total protein	−0.14	**0.75**	0.26
Thaw drip loss	**0.83**	−0.04	0.09
Thaw drip loss protein	−0.47	−0.00	−0.22
WHC_raw_	−0.33	0.66	−0.25
WHC_cook_	−0.63	−0.14	**−0.66**
WHC_total_	−0.40	−0.08	**−0.79**
TVB-N	−0.53	−0.62	−0.16
pH	**0.74**	−0.27	−0.03
Total phosphates in fillets	**0.76**	0.55	−0.02
Total phosphates in thawing waters	**0.81**	0.43	−0.26
Orthophosphates in fillets	**0.79**	0.53	−0.10
Orthophosphates in thawing waters	0.49	0.66	−0.47
Overall score of negative descriptors of raw fillets	0.56	−0.48	−0.46
Overall score of negative descriptors of cooked fillets	**0.76**	−0.31	−0.32
Overall score of positive descriptors of cooked fillets	**−0.83**	0.35	0.19

**Table 10 foods-10-00848-t010:** Linear correlation between objective (M/P ratio, protein, soluble protein, pH, thaw drip loss, WHC_raw_, WHC_cook_, and storage length) and subjective (overall quality of cooked fillets) determinations. R, regression coefficient; *p*-value, significance level.

Variables	R	*p*-value	Estimated Limits *	95% Confidence Limits
M/P ratio vs. overall quality	−0.810	<0.01	**4.3**	± 0.2
Protein vs. overall quality	0.808	<0.01	**18.2%**	± 0.5
Soluble protein vs. overall quality	0.805	<0.01	**12.2%**	± 0.4
pH vs. overall quality	−0.691	<0.05	**6.7**	± 0.1
Thaw drip loss vs. overall quality	−0.657	<0.05	**3.4%**	± 3.4
WHC_raw_ vs. overall quality	0.563	<0.05	**55.7%**	± 2.6
WHC_cook_ vs. overall quality	0.563	<0.05	**59.1 %**	± 1.5
Storage length vs. overall quality	−0.556	<0.05	**6.8 months**	± 3.9

* Estimated values calculated for an overall sensory quality of cooked products ≥ 3.

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
