# Peer review of "Quality of Frozen Hake Fillets in the Portuguese Retail Market: A Case Study of Inadequate Practices in the European Frozen White Fish Market"

_foods, 2021, doi:10.3390/foods10040848_

Round 1
Reviewer 1 Report
The manuscript describes a study of various quality parameters, including species detection by PCR, sensory evaluation, added phosphate, protein loss and microbiological analysis in frozen Hake fillets sampled from the Portuguese retail marked. The study is interesting and relevant, and thoroughly discussed. However, the quality assurance for the analysis should be better explained, and the manuscript needs to be improved with parts clarified and explained, as well as the language needs to be improved.
Please see the specific comments given below.
In this study the quality of frozen hake has been evaluated by determining the moisture content, protein content, soluble proteins, polyphosphates, citrate, as well as hake species identification and microbiological
analysis. However, it is not conducted other biochemical analysis, such as lipid content, fatty acids, ash and TBARS. Please explain the reason for the selected parameters hereby studied, and why other parameters
has not been included. Please also make this clear in the manuscript.
In the introduction of the manuscript, there is quite extensive information on fish consumption and hake in Portugal/Europe, whereas important topics, e.g. a description of the regulations in Portugal and Europe in
terms of quality of food and labelling, is not well introduced. Furthermore, information on the what parameters needs to be evaluated, and how these are commonly evaluated is lacking. Also, an introduction
of the topics you are here studying, such as the species identification with PCR, proteins, phosphate, M/P ratio, and the microbiological evaluations are not introduced for the reader. In the results and discussion of
the manuscript, there is on the other hand extensive information for each analysis/parameter, and some of this information could be moved to the introduction of the manuscript. In this way, the result and discussion
would also be easier to follow.
In the material and method section, the analysis conducted are described well with references to methods/standards applied. However, there is missing information on the quality assurance for the measurements. Please include details about the quality control for the measurements. Furthermore, are the sensory evaluation controlled in some way?
“A case study of inadequate practices in the European frozen fish market” is used as a part of the title of the manuscript. Does your data justify this strong statement? It is stated in the result section (Table 6) that the
overall quality of 79% of the raw fillets are evaluated as excellent and good. Is this correct? For the cooked fish, 63% of the samples are evaluated to have poor or bad quality (Table 7). The differences in evaluation of fresh and cooked fish fillets are not well discussed. and how would the quality evaluation of fresh fish be taken into consideration for the overall conclusion?
In page 6, line 218-220, it is stated that the fish is steam-cooked for 10 min at 100 C, whereas for the driploss (Page 4, Line 125-126), the fish is steam-cooked for 15 min. Why are these different, and what is preferred temperature and time for cooking of hake? Has this been evaluated?
Page 3; Line 94: is 13 packages from each brand (20x13= in total 260 samples) being analysed for all methodologies, or is pooled samples used? Please specify in the description for the analysis under the materials & methods section, and importantly also in the tables, what n is equal to.
Page 13, Line 537, and Page 14, Line 538-539: It is stated that the total phosphate levels are overestimated.
What is meant with this? Please make it clear in the manuscript (either M&M or R&D section) how much overestimated these measurements are, and how this is controlled.
An overall comment to the result & discussion: there is generally a high number of references to studies by others, both in the start and then end of some of the sections, and because of this your results are not always
clear. Please make your results being the most important part, being clearly stated. Your reference list is also quite long and could probably be shortened.
The conclusion of the study is long and should preferably be shortened and becoming more concise.
Table 4: A reference to studies by others is given as a footnote. Please move this to the discussion of the
results.
Table 6: is the frequency expressed as %? Please specify what the frequency is.
Page 2, Line 51-55: Please include references to these statements.
Page 2, Line 86: please rephrase “the universe of brands” into e.g. “the number of brands”. How many brands were detected in the survey? Please specify the type of samples collected, being fish fillets.
Page 4, Line 112: please include the brand of the refrigerated chamber.
In page 5, Line 188: please make it clear what microbiological variables (bacteria, mold, etc) that the fish has been analysed for.
Page 12, Line 444: it is not clear to me what is meant with “succulence quotations”. Please change the wording or explain to the reader.
Page 13, Line 529-530: It is stated that phosphate levels are correlated with the contents in the fillets. What content? Please, make it clear what the phosphate levels are correlated with.
Page 14, Line 549: Please include the detector used as well, and not just the chromatography.
Table 5: Please change the e.g. “<0.02” with “<LOQ” or “<LOD”, and these abbrevations should also be explained in the table text.
Page 14, Line 558: when stating “not quantifiable” then the LOQ (limit of quantification) should be stated.
Page 15, Line 587-588: Please remove “in our laboratory” in this sentence. Also, remove this type of wording in other parts of the manuscript as well. How many samples are there (n=?), and if this is a control
sample for the analysis, these details should be included in the Materials and Methods section of the manuscript.
Page 16, Line 655: Please use either only “58%” or “most frequently found” (not “most frequently found in
58%”).
Several of the sentences in the manuscript are long and difficult to follow. I would recommend that a native English speaker correct for spelling errors.
Author Response
REVIEWER 1
Title: The quality of frozen hake fillets in the Portuguese retail market: A case study of inadequate practices in the Europeans frozen white fish market
We wish to express our gratitude to the reviewers for their insightful comments and thorough review, which have helped us significantly improve the paper and also to the editor for the expeditious processing time.
In this letter, as suggested, replies to reviewers’ comments are addressed point–by-point and reported in Italics.
In the revised manuscript the changes are tracked.
Supplementary materials are also provided.
Thank you for your kind attention
Best regards
Comments to the authors
The manuscript describes a study of various quality parameters, including species detection by PCR, sensory evaluation, added phosphate, protein loss and microbiological analysis in frozen Hake fillets sampled from the Portuguese retail marked. The study is interesting and relevant, and thoroughly discussed. However, the quality assurance for the analysis should be better explained, and the manuscript needs to be improved with parts clarified and explained, as well as the language needs to be improved.
Please see the specific comments given below
In this study the quality of frozen hake has been evaluated by determining the moisture content, protein content, soluble proteins, polyphosphates, citrate, as well as hake species identification and microbiological analysis. However, it is not conducted other biochemical analysis, such as lipid content, fatty acids, ash and TBARS. Please explain the reason for the selected parameters hereby studied, and why other parameters has not been included. Please also make this clear in the manuscript.
REPLY: We would like to thank the reviewer for the remark. Regarding the criteria used in the selection of the biochemical analysis performed it was based on the ability to better reflect the level of performance of the fish industry in the supply of hake fillets to the consumers and therefore, that could better evidence the quality changes occurring during processing and frozen storage and also compliance of products with labelling regulations. Other biochemical analysis such as the mentioned (lipid content, fatty acids, ash and TBARS) were not considered relevant, mainly in the case of the lipid fraction, because hake is a lean species and most of the changes ascribed to freezing and frozen storage are caused by loss of textural attributes rather than oxidation. Nevertheless, oxidation of lipids was followed by sensory evaluation of raw and cooked products.
Authors clarified these aspects in the text.
Line 168: Selection of the biochemical analysis performed was based on the ability to better evidence the quality changes occurring during processing and frozen storage and also the compliance of products with labelling regulations. Other biochemical analysis namely in the lipid fraction, were not considered because hake is a lean species and most of the changes ascribed to freezing and frozen storage are caused by loss of textural attributes rather than oxidation.
In the introduction of the manuscript, there is quite extensive information on fish consumption and hake in Portugal/Europe, whereas important topics, e.g. a description of the regulations in Portugal and Europe in terms of quality of food and labelling, is not well introduced. Furthermore, information on the what parameters needs to be evaluated, and how these are commonly evaluated is lacking. Also, an introduction of the topics you are here studying, such as the species identification with PCR, proteins, phosphate, M/P ratio, and the microbiological evaluations are not introduced for the reader. In the results and discussion of the manuscript, there is on the other hand extensive information for each analysis/parameter, and some of this information could be moved to the introduction of the manuscript. In this way, the result and discussion would also be easier to follow.
REPLY: we would like to thank the reviewer for the precious insight. The regulatory framework in Portugal and Europe in terms of quality of food and labelling has been added to Introduction as well as information regarding the quality parameters used in the hake fillets evaluation.
Line 106-142: Regulatory framework in Portugal and Europe
In the material and method section, the analysis conducted are described well with references to methods/standards applied. However, there is missing information on the quality assurance for the measurements. Please include details about the quality control for the measurements. Furthermore, are the sensory evaluation controlled in some way?
REPLY: we would like to thank the reviewer for the comment. The analysis were performed in an accredited laboratory according to ISO 17025:2017. TVB-N, Moisture, Salmonella and E. coli analysis are accredited determinations (PORTUGUESE ACCREDITATION INSTITUTE - Technical Annex LO258-1). The other determinations followed the same type of internal, and external quality assurance, namely targeted to the internal control of the precision and/or errors in the performance of the methods (e.g. analysis of duplicates, control standards, internal reference materials, verification of calibration curves) and external control of the accuracy of results (e.g. analysis of certified reference materials, recovery and proficiency tests/interlaboratory comparisons). Sensory analysis was controlled in terms of repeatability by using a fixed panel of 10 trained assessors.
Authors clarified these aspects in the text.
Line 270: In quality assurance of measurements a maximum relative error of 5% was, in general, accepted for validation of methods precision and accuracy.
“A case study of inadequate practices in the European frozen fish market” is used as a part of the title of the manuscript. Does your data justify this strong statement? It is stated in the result section (Table 6) that the overall quality of 79% of the raw fillets are evaluated as excellent and good. Is this correct? For the cooked fish, 63% of the samples are evaluated to have poor or bad quality (Table 7). The differences in evaluation of fresh and cooked fish fillets are not well discussed. and how would the quality evaluation of fresh fish be taken into consideration for the overall conclusion?
REPLY: we appreciate the call for attention. We tried to summarize in the tittle the unacceptable amount of deficiencies that were found at different levels in the studied products. With technological development inducing in the fish industry more advanced processing and storage procedures, studies in frozen fish have lost frontpage relevance in the last years, possibly because it was assumed that the industry (producers/retailers) were complying with best practices. The title is for that reason a call for attention and is in great extent due to the poor sensory quality of the cooked products. However, it is not unexpected that the evaluation of raw products originates a higher percentage of acceptable products than the evaluation of cooked products. However, the high level of acceptance of the raw fillets does not invalidate the conclusions. Since the fish fillets are consumed cooked it were ultimately the attributes from this particular sensory experience and the characteristics that consciously or unconsciously the fish eater (test panel) considers should be present, that were taken into consideration in detriment of the raw products evaluation.
In fact, in sensory evaluation the panel evaluated the raw products as perceived through the 2 senses - sight, and smell, whereas in the evaluation of cooked products taste evaluation was also added. Taking into consideration that in sensory evaluation of cooked fish heat induces the volatilization of specific organic compounds enhancing smell and taste perception, this makes more detectable the degradation and oxidation flavours presented in the products and justifies the raw/cooked differences.
Authors clarified these aspects in the text.
Line 847: Comparison of raw and cooked sensory evaluation showed considerably lower rating in cooked fillets. This difference is mainly explained by the contribution of taste scoring in cooked fillets and the effect on this of the volatilization of specific organic compounds induced by heat. Cooking enhances the smell and taste perception of these molecules and makes more detectable the degradation and oxidation flavours presented in the products.
Since the fish fillets are consumed cooked it were ultimately the attributes from this particular sensory experience and the characteristics that consciously or unconsciously the fish eater (test panel) considers should be present in a cooked product, that were taken into consideration for overall conclusion in detriment of the raw products evaluation. In summary, no product was rated as good or excellent,…
In page 6, line 218-220, it is stated that the fish is steam-cooked for 10 min at 100 C, whereas for the drip- loss (Page 4, Line 125-126), the fish is steam-cooked for 15 min. Why are these different, and what is preferred temperature and time for cooking of hake? Has this been evaluated?
REPLY: We would like to thank the reviewer for the remark and we understand that the reviewer was mentioning cook loss.
Temperature (100 ºC) and time (10 min) of hake steam cooking for sensory analysis is a standard operating procedure in the Sensory Laboratory. This specific procedure has resulted from experimental adjustment that took into consideration the size of the sample (individual portions), the type of wrapping (aluminium foil), the level of development of volatiles and preservation of odours, the effect in the integrity of the muscle and the preservation of the flavours, among other.
For Cook Loss there is no standard operating procedure and some referenced methods indicate 5 min [59], 15 min [58] or 20 min [62]. The method used in this study involved wrapping the all fillet in polyamide and polyethylene film bags and vacuum packaging. This, because of the higher mass of these samples in relation to sensory analysis samples and also the higher heat transference of the aluminum foil we used for Cook loss a higher cooking time for attaining a similar time/temperature in the core as in the samples for sensory analysis.
Page 3; Line 94: is 13 packages from each brand (20x13= in total 260 samples) being analysed for all methodologies, or is pooled samples used? Please specify in the description for the analysis under the materials & methods section, and importantly also in the tables, what n is equal to.
REPLY: we appreciate the call for attention regarding sampling description. For analysis 13 packages from each brand and batch were sampled, from these 10 packages were used for determination of packaging analysis, net weight, glaze ice, and sensory analysis and 3 packages (pooled) used in the other determinations. Text was edited in order to clarify this issue and n= added in the tables legends.
Line 191: A total of 13 packages of the same brand and batch were collected in October 2018 in the selected food shops, transported to the laboratory under controlled temperature (-18 oC) and kept stored at -20°C ± 1oC until analysis within 1 week. From these, 10 packages were used for analysis of packaging (packaging defects, presence of ice and fillet’s size variation), net drained weight, glaze ice, and sensory analysis and 3 packages (pooled) used in the other determinations. Individual weight of hake fillets packages ranged from 0.4 kg to 1.0 kg and no records of storage temperatures were obtained from processors or retailers.
Page 13, Line 537, and Page 14, Line 538-539: It is stated that the total phosphate levels are overestimated. What is meant with this? Please make it clear in the manuscript (either M&M or R&D section) how much overestimated these measurements are, and how this is controlled.
REPLY: The spectrophotometric method consists in the quantification of total phosphorus, and the result is then converted to phosphates contents. The overestimation authors mentioned was considering phosphorus that is not in the form of phosphates (e.g. inorganic ions, phosphoproteins, phospholipids, ATP, DNA). In fact, for the case of fillets, that contribution (e.g. due to the presence of a fish bone/spines) could be neglected, and for that reason, the sentence “Furthermore, quantification of total phosphates is overestimated since it is based in the conversion of total phosphorus in phosphates expressed as P2O5.” was deleted from the manuscript.
An overall comment to the result & discussion: there is generally a high number of references to studies by others, both in the start and then end of some of the sections, and because of this your results are not always clear. Please make your results being the most important part, being clearly stated. Your reference list is also quite long and could probably be shortened.
The conclusion of the study is long and should preferably be shortened and becoming more concise.
REPLY: we appreciate the call for attention and text was edited in order to follow the reviewer suggestion.
Table 4: A reference to studies by others is given as a footnote. Please move this to the discussion of the results.
REPLY: We would like to thank the reviewer for the remark. The reference was moved to discussion according to reviewer suggestions and the text changed.
Line 546: Samples of this species displayed also the highest M/P ratio, 5.5 ± 0.2 (Table 4). Reported M/P ratio have been accounted as 4.7 ± 0.2 in M. capensis/paradoxus, 4.8 ± 0.2 in M. hubbsi and 5.0 ± 0.2 in M. productus [85].
Table 6: is the frequency expressed as %? Please specify what the frequency is.
REPLY: We would like to thank the reviewer for the remark. The frequency distribution of attributes and overall quality scores refers to the number of times the different sensory scores occur in the sensory evaluation of the 19 commercial hake fillet products. By presenting the data in this way instead of a table with all the individual scores of each sample, the results are more informative and it is possible a better evaluation of the overall performance of the suppliers in regard to the different attributes.
Page 2, Line 51-55: Please include references to these statements.
REPLY: We would like to thank the reviewer for the remark. Reference was added according to the reviewer suggestions.
Line 1023: [10] – Moody M.W. Fish Processing. In: Encyclopedia of Food Sciences and Nutrition (2nd Ed.), Ed. Trugo, L; Finglas, P.M., Academic Press,2003, pp. 2453-2457
Page 2, Line 86: please rephrase “the universe of brands” into e.g. “the number of brands”. How many brands were detected in the survey? Please specify the type of samples collected, being fish fillets.
REPLY: We would like to thank the reviewer for the remark. The survey detected 32 brands in 13 points of sale. The 20 brands with the higher frequency of presence in the retailers were sampled. Text was added according to reviewer suggestions.
Line 187: The survey detected a total of 32 hake fillet brands from which the 20 brands with the higher frequency of presence in the retailers were selected (Table 1).
Page 4, Line 112: please include the brand of the refrigerated chamber.
REPLY: Reference of the refrigerator was added according to reviewer suggestions
Line 217: Fiocchetti Labor 500 ECT-F (Luzzara, Italy)
In page 5, Line 188: please make it clear what microbiological variables (bacteria, mold, etc) that the fish has been analysed for.
REPLY: We would like to thank the reviewer for the remark. Text was added according to reviewer suggestions.
Line 295: Microbiological testing involved enumeration of total aerobic microorganisms, sulphite-reducing bacteria, total coliforms and E. coli, yeast and mold, Staphylococcus aureus and detection of Salmonella Sp.
Page 12, Line 444: it is not clear to me what is meant with “succulence quotations”. Please change the wording or explain to the reader.
REPLY: We would like to thank the reviewer for the remark. Succulence quotations refers to the scores given by the test panel to the succulence attribute of the fish fillets.
The text was changes for clarification.
Line 847:… of the correlation (p<0.01) of succulence scores.
Page 13, Line 529-530: It is stated that phosphate levels are correlated with the contents in the fillets. What content? Please, make it clear what the phosphate levels are correlated with.
REPLY: The text was edited for clarification that the correlation is between total phosphates in thawing water and total phosphates in fillets.
Line 670: Results ranged between 2.1 g and 5.1 g P2O5/L and showed a significant correlation (p<0.01) with the total phosphates contents in the fillets.
Page 14, Line 549: Please include the detector used as well, and not just the chromatography.
REPLY: The type of detector (Dionex conductivity detector) is identified in the Material and Methods, but the authors further clarified these aspects in the text.
Line 258: …with conductivity detection (IEC)…
Table 5: Please change the e.g. “<0.02” with “<LOQ” or “<LOD”, and these abbreviations should also be explained in the table text.
REPLY: Table 5 was modified according to reviewer suggestions and abbreviations explained in the table footnote.
Line 704: Table 5
Page 14, Line 558: when stating “not quantifiable” then the LOQ (limit of quantification) should be stated.
REPLY: Text was modified according to reviewer suggestions
Line 689: In the case of diphosphate (P2O7), levels in the fillets were in general lower than the limit of detection (<0.02 g P2O5/kg) with exception of sample #15 that showed traceable amounts, though not quantifiable (LOQ=0.05 g P2O5).
Page 15, Line 587-588: Please remove “in our laboratory” in this sentence. Also, remove this type of wording in other parts of the manuscript as well. How many samples are there (n=?), and if this is a control sample for the analysis, these details should be included in the Materials and Methods section of the manuscript.
REPLY: The sentence was modified accordingly to reviewer suggestions as well as in other parts of the manuscript. The controlled sample was not mentioned on the Material and Methods because is not a control sample of this study (analyzed at the same time as the fillet samples). The term controlled refers to the fact that its origin in research vessels warranties that is controlled and that it was not submitted to any kind of processing.
The sentence was modified accordingly to avoid confusion with control samples.
Line 726: Similar results (<LOQ) were previously obtained in controlled samples of Merluccius merluccius from the Portuguese coast not processed with citrates (data not published).
Page 16, Line 655: Please use either only “58%” or “most frequently found” (not “most frequently found in 58%”).
REPLY: The sentence was modified accordingly to reviewer suggestions.
Line 798:… being a period between 23-27 months the most frequently found.
Several of the sentences in the manuscript are long and difficult to follow. I would recommend that a native English speaker correct for spelling errors.
REPLY: We would like to thank the reviewer for the remark. A review of the spelling errors and a simplification of the explanation was done, hopefully with success.
Reviewer 2 Report
The knowledge of the overall quality of frozen hake fillets in the Portuguese market as a case study is very informative for industry, regulatory authorities and the public in general, and it could help in improving manufacturing and storage practices, as well as rising the overall culture of the public in the type of food products they are purchasing.
The results presented in this paper are interesting although more precision in some methodological aspects is needed. In addition, given the importance of the sensory analysis in the overall picture presented in this piece of work, both, materials and methods and results should be more clearly described and presented. Moreover, authors should critically revise the results based on hedonic scales, which are not probably precise enough for a study of these characteristics, or at least for drawing such strong conclusions. Please note that they used experienced panellists and probably their appreciation of the overall quality of seafood is significantly more exquisite that that of the average consumer. There are well defined sensory scales for cooked white fish and the sensory panel could objectively use.
The description of the overall changes occurring in frozen hake upon storage is described in a fragmented way and sometimes it looks even contradictory thoughout the text. It is suggested that authors make an effort to avoid this fragmentation. Some thresholds are given for certain values/indexes but they not always seem to be justified, and they seem to be made “ad hoc”. Many of them should be agreed upon the sector and authorities, and used only if they gave scientifically relevant results. In this sense, this reviewer thinks that the authors have an excellent opportunity to discriminate between informative indicators from those which may be secondary in this context. The conclusions are in some aspects based on opinions rather than in facts. Occasionally, results are presented in the conclusions without being mentioned before.
The title is very catching indeed, but I believe that authors should give it a second thought. It is directed to the whole Sector without any discrimination between processors who may be doing fine (or complying with legislations) from those that are not. Another issue is that the retail shops may store their goods in a way that the distinct characteristics of products that had processed with good practices are rapidly lost. No information on this factor as regards the overall indicators has been given. For example, were all the retail shops complying with the storage temperature? Was this temperature different? Did this have an effect on the final values? How replications from the same brand were purchased in different shops?
These are examples for the overall impression of this manuscript. I suggest authors perform a thorough revision before the paper is ready for publication.
More specific points
Introduction
It is suggested that authors define from start which do they mean by quality and which are the criteria to measure this quality.
Lines 49-50: The sentence “quality issues in frozen fish are mostly related to dehydration and oxidation” is too vague and incomplete. I suggest to describe here (from start) all the aspects related to the loss of muscle characteristics that lead to loss of lean fish quality, rather than little by little throughout the text. Please note that hake is a lean species and most of the changes attributed to freezing and frozen storage are due to loss of textural attributes rather than oxidation. Products become tough, dry and with high expressible moisture loss. This involves denaturation of myofibrillar proteins but mainly aggregation. This aspect is scarcely mentioned in the text.
Materials and methods
The sampling plan should be more clearly specified: number of shops where samples were purchased, if they replicated the same brand in different shops, if they had data on the storage conditions in each shop. Some of the information contained in the packages is missing. Please note that authors refer to some issues in the text but there are not presented as tables (i.e. nutritional labelling, etc).
The time elapse between the purchase and the actual analysis is relevant in this context and it should be stated. Please note that if authors stored their samples at -18 ºC, unless all the analyses were performed in a very short time, chances are that the laboratory storage itself might have an impact on the results. Authors state that according to one of the indicators, the average storage time/temperature combination was 3 months at -18 ºC. At the laboratory storage temperature they use, a few weeks of delay in the analyses may have a significant impact on the results. I suggest including this factor in the discussion.
As stated before, more details on the sensory analysis should be given: For example, the integrity of the package was evaluated with a hedonic scale: how was that described? How many assessors? Results of this indicator have not been presented. Authors should reconsider shifting this part included in point 2.2.1 (i.e. physical analysis) to point 2.2.5 (sensory analysis).
Which was the size of the samples to be cooked? (lines 123-126).
Please indicate if in the determination of soluble protein the non-protein nitrogen content was substracted (lines 136-137).
No information on the number of packages analysed per brand in each of the indicators is found. Did authors sampled all packages for all brands? Perhaps a sampling plan could help here (a diagram for example).
Line 163: Please indicate in which units are results expressed.
Line 214-216: It is not clear how the overall quality determination was performed. Please clarify.
Lines 231-237: Please specify which quality parameters were used in PCA.
Results and discussion
Table 3: The column “net weight difference” does not seem to reflect this. If understood properly, it is calculated as a percentage of actual vs declared weigh. Please rephrase.
Line 273: …”which the need to adjust a specific net weight at packaging does not justify”. The sentence seems weird. Please rephrase.
Lines 275-278: the statement “…showing glazing levels higher than the normal range used in the industry” seems contradictory with authors´ results. Actually they show that the actual levels used in industry range between 5.8-32.9%. It is true that their results contrast with those of data from reference 40. But this may show one part of the industry whereas authors´ data show another part. I suggest to rephrase the paragraph.
There are several grammar errors that should be corrected. For example, Line 291: Replace “restores” by “restore”.
Lines 296-298: The sentence: “Processes occurring during frozen storage, like protein surface dehydration or, to a minor degree, physical damage in the membranes or cells have been considered the causative processes of this decrease” (i.e. water holding capacity). This is an example of descriptions that should described at once and in a more precise way.
Lines 299-303: Thaw drip loss of samples 9, 15, and 16 may arise from water addition to Merluccius productus. Rough calculation (i.e. sample number 9) suggest that, for average moistures of 80% for hake, given that moisture of this sample is about 83% (table 4) there was about 90 g of added water in 600 g of product. This would mean that about 15 g out of the 25 g of drip loss may come from added water. I suggest to think around these lines since chances are that rather than having an species-difference in quality, the differences observed in M productus are, in this case, due to this factor. Please note that these are the three samples that have some traces of phosphate and according to authors, they come from the same producer. If this is true, then authors could revise their discussion.
Lines 401-402: Correlations are mentioned between pH and soluble protein but the latter has not been described yet. Please reorder this type of statements here and throughout the text.
Lines 435-437: Please consider if the unexpectedly high values authors find are due to non protein nitrogen. If this is not separated from the protein soluble fraction (i.e. by precipitation of soluble proteins with TCA or PCA for example) this nitrogen may be erroneously considered protein nitrogen.
Lines 460-462: The practical storage time in the frozen state increases considerably by lowering the storage temperature. -18 ºC is too high. Authors may want to include this point in the discussion of this part. At least some industries keep their products well below 18 ºC so there may not necessarily by incorrect practices (i.e. mislabelling).
Section 3.5. Total Volatile Nitrogen is mainly used for monitoring microbial changes during storage temperatures above 0 ºC. Because hake is a formaldehyde-forming species, authors could have chosen to analyse TMA and DMA, rather than TVBN. Increase of DMA [and formaldehyde (FA)] values are relevant at -18 ºC but the reaction halts at -30 ºC. It is well known that FA has a detrimental effect resulting in tough products owing to the aggregation of muscular proteins.
Line 625: Authors state that there is no microbiological problem since hake fillets are eaten after thoroughly cooking. This is an opinion. They should state if this is within the accepted levels, since they mention that the presence of moulds reflects poor hygienic conditions. A table with the microbiological data per brand would be of help.
Lines 640-642: Hake suffers mainly aggregation.
The actual practical storage time depends, all the rest of factors being constant, on the time but also the temperature of storage. Thus if samples have been stored at -25 ºC (some industries do this), two years may be adequate. I suggest to consider this in the discussion.
I suggest to include tables with sensory analysis of raw and cooked hake fillets with information by brand, as it was done in the former analyses (i.e. tables 1-5).
In table 6 it is not clear how the overall quality has been obtained. It seems strange that if there is 5% of the brands (i.e. one out of 20) displaying score 5 (Bad) in 5 out of 7 descriptors, the overall quality for a bad sample is 0%. Of course, chances are that there is not always the same brand having these bad scores, but it should be better explained.
Table 7 is difficult to follow since the positive and negative scores have the same scale (i.e. absent=1) but meaning completely different things, and are on the same row, and thus it is difficult to get an idea of what is happening. Moreover, Overall quality (i.e. 1, bad) is on the same row of rancid (also score 1, absent). I suggest to change the laying out of the table and make it easier to read. Again, this reviewer suggest to include tables with the actual values of each descriptor per brand. This would help the reader relate the biochemical with sensory values more easily.
What does firmness (1-absent) mean?
Figure 1, replace PCA with PC.
Line 706: lower PCA1 values is meaningless. Please rephrase thoughout the text.
Line 709: from this statement, it looks that PCA has been performed also with sensory values. Please explain which variables were used in the analysis.
Please note that results from fig1 may not be species related. As explained before, they may be due to the fact that samples 9, 15 and 16 are from the same processor and they all (may) have added water. (Fig 1 and lines 721-726).
As regards factor analysis, please consider including a loading table, so that readers can see which are the variables that entered in each of the PC. Also, please indicate which is the % of explained variance for each PC.
Conclusions
As stated before, the conclusions are in some aspects based on opinions rather than in facts. For example, in lines 775-776, one may argue on what bases the quality has been defined as poor to medium. I suggest to be more specific and differentiate between what is mandatory by regulation (criteria and threshold levels), in which criteria there is a wide scientific consensus for a given threshold, and what it is a suggestion of the authors.
Occasionally, results are presented in the conclusions without being mentioned before. For example, in lines 777-779, authors mention that fillets from M. paradoxus were the most appreciated by the sensory panellists…etc but this was not presented before.
Lines 780-784, please see the previous comment on the cooked samples/microbial analyses.
Line 791: ‘best before’ longer than recommended… by whom? I strongly suggest to look carefully at the conclusions.
Line 794: The sentence: “No product has been rated as good or excellent by a trained sensory panel”. Hedonic scales are usually done for consumer studies but with a much higher number of persons. Chances are that results are biased so that they not reflect the view of the average consumer.
I suggest Allan Bremner´s paper (Toward Practical Definitions of Quality for Food Science, 2000, Critical Reviews in Food Science and Nutrition 40(1):83-90). He defines an approach which links the concept of quality, through a general definition, by adding the “missing link of specific definitions related to measurable attributes and properties determined by standard methods to provide values that can be used to evaluate foods or to set specifications”.
Author Response
REVIEWER 2
Manuscript number: foods-1137891
Title: The quality of frozen hake fillets in the Portuguese retail market: A case study of inadequate practices in the Europeans frozen white fish market
We wish to express our gratitude to the reviewers for their insightful comments and thorough review, which have helped us significantly improve the paper and also to the editor for the expeditious processing time.
In this letter, as suggested, replies to reviewers’ comments are addressed point–by-point and reported in Italics.
In the revised manuscript the changes are tracked.
Thank you for your kind attention
Best regards
Comments and Suggestions for Authors
The knowledge of the overall quality of frozen hake fillets in the Portuguese market as a case study is very informative for industry, regulatory authorities and the public in general, and it could help in improving manufacturing and storage practices, as well as rising the overall culture of the public in the type of food products they are purchasing.
The results presented in this paper are interesting although more precision in some methodological aspects is needed. In addition, given the importance of the sensory analysis in the overall picture presented in this piece of work, both, materials and methods and results should be more clearly described and presented. Moreover, authors should critically revise the results based on hedonic scales, which are not probably precise enough for a study of these characteristics, or at least for drawing such strong conclusions. Please note that they used experienced panellists and probably their appreciation of the overall quality of seafood is significantly more exquisite that that of the average consumer. There are well defined sensory scales for cooked white fish and the sensory panel could objectively use.
The description of the overall changes occurring in frozen hake upon storage is described in a fragmented way and sometimes it looks even contradictory thoughout the text. It is suggested that authors make an effort to avoid this fragmentation. Some thresholds are given for certain values/indexes but they not always seem to be justified, and they seem to be made “ad hoc”. Many of them should be agreed upon the sector and authorities, and used only if they gave scientifically relevant results. In this sense, this reviewer thinks that the authors have an excellent opportunity to discriminate between informative indicators from those which may be secondary in this context. The conclusions are in some aspects based on opinions rather than in facts. Occasionally, results are presented in the conclusions without being mentioned before.
REPLY: we appreciate the opinion of the reviewer. Fragmented information regarding the quality deterioration process was consolidated in the Introduction. The regulations and conformity parameters were also consolidated in the Introduction. Thresholds for some parameters are based on bibliographic data and are not accounted in terms of conformity. Others are proposed by the authors based on the results obtained, but these are merely suggestion because of the uncontrolled nature of the data and the need for an increased size of the sample for confirmation, due to the inferior level of significance of the correlations found (p<0.05).
The title is very catching indeed, but I believe that authors should give it a second thought. It is directed to the whole Sector without any discrimination between processors who may be doing fine (or complying with legislations) from those that are not. Another issue is that the retail shops may store their goods in a way that the distinct characteristics of products that had processed with good practices are rapidly lost. No information on this factor as regards the overall indicators has been given. For example, were all the retail shops complying with the storage temperature? Was this temperature different? Did this have an effect on the final values? How replications from the same brand were purchased in different shops?
REPLY: we appreciated the overall opinion and suggestions of the reviewer and in recognition a considerably effort was dedicated into following the recommendations. It is clear for us that there are business operators that perform under the best practices and complying with legislation, both in the processing industry and in the retail sector, as it is evidenced by the good quality of some of the products. To account for that and make it clearer we changed our perspective and therefore address the call for attention not to the whole sector but to a significant part of the hake fillets business operators in the fishery sector (e.g., processors, retailers). However, because the results of this and past studies have been so disappointing in regard the majority of the brands, we feel that strong messages should be produced in order to change the present situation in benefit of consumers and to defend business operator that deliver good quality products. To obtain more details and determine if the problems are in the quality of the raw materials, in the processor’s practices or in the retailer’s storage conditions was outside of the scope of the present study, which was mostly focused on the quality of the products that reach consumers and determination of possible problems. To do a vertical audit of all business operators to ascertain the “week links”, though extremely interesting faces unsurmountable difficulties because of non-access to business operator information, with the exception of the retailer’s freezers that is this case were all at the reglementary temperature (-18 oC).
These are examples for the overall impression of this manuscript. I suggest authors perform a thorough revision before the paper is ready for publication.
More specific points
Introduction
It is suggested that authors define from start which do they mean by quality and which are the criteria to measure this quality.
Lines 49-50: The sentence “quality issues in frozen fish are mostly related to dehydration and oxidation” is too vague and incomplete. I suggest to describe here (from start) all the aspects related to the loss of muscle characteristics that lead to loss of lean fish quality, rather than little by little throughout the text. Please note that hake is a lean species and most of the changes attributed to freezing and frozen storage are due to loss of textural attributes rather than oxidation. Products become tough, dry and with high expressible moisture loss. This involves denaturation of myofibrillar proteins but mainly aggregation. This aspect is scarcely mentioned in the text.
REPLY: we appreciate the reviewers’ comments and helpful suggestions. Introduction was edited in order to accommodate “all the aspects related to the loss of muscle characteristics that lead to loss of lean fish quality”.
Line 61-105: Aspects related to the loss of muscle characteristics that lead to loss of lean fish quality
Line 106-142: Aspects related to applicable regulations
Materials and methods
The sampling plan should be more clearly specified: number of shops where samples were purchased, if they replicated the same brand in different shops, if they had data on the storage conditions in each shop. Some of the information contained in the packages is missing. Please note that authors refer to some issues in the text but there are not presented as tables (i.e. nutritional labelling, etc).
REPLY: we appreciate the request for clarification. The sampling description was edited in order to add the information suggested. Regarding the reviewer comment “Some of the information contained in the packages is missing” we stated in the manuscript that “it is notable that all labels respect legal requisites (data not shown)”. Thus, to avoid overloading the manuscript with information that would be redundant that information was not presented. Since other information (e.g. nutritional and energy information, how thawing and/or preparation of product should be done) is voluntary we do not present it in a table also to not overload the article, but mention its importance to consumers as an opportunity for improvement.
Line 184: In order to identify the number of brands of deep-frozen pre-packaged hake fillets available in the Portuguese retail market, a survey was carried out in 13 small traditional food shops (minimarkets, traditional markets and frozen fish shops) and large food retail chains (hyper/supermarkets). The survey detected a total of 32 hake fillet brands from which the 20 brands with the higher frequency of presence in the retailers were selected for analysis (Table 1). Given the growing importance of own brands of large food retail chains, about half of the sample is composed of these brands (45% large food retail chains brands and 55% manufacturer brands). A total of 13 packages of the same brand and batch were collected in October 2018 in the selected food shops, transported to the laboratory under controlled temperature (-18 oC) and kept stored at -20°C ± 1oC until analysis within 1 week. From these, 10 packages were used for analysis of packaging, net weight, glaze ice, and sensory analysis and 3 packages (pooled) used in the other determinations. Individual weight of hake fillets packages ranged from 0.4 kg to 1.0 kg and no record of the storage temperature was obtained from retailers.
The time elapse between the purchase and the actual analysis is relevant in this context and it should be stated. Please note that if authors stored their samples at -18 ºC, unless all the analyses were performed in a very short time, chances are that the laboratory storage itself might have an impact on the results. Authors state that according to one of the indicators, the average storage time/temperature combination was 3 months at -18 ºC. At the laboratory storage temperature they use, a few weeks of delay in the analyses may have a significant impact on the results. I suggest including this factor in the discussion.
REPLY: we appreciate the request for clarification. The sampling description was edited in order to add the information suggested. Given the short storage period at the laboratory this was not accounted in the discussion of the results.
Line 191: A total of 13 packages of the same brand and batch were collected in October 2018 in the selected food shops, transported to the laboratory under controlled temperature (-18 oC) and kept stored at -20 ± 1oC until analysis within 1 week.
As stated before, more details on the sensory analysis should be given: For example, the integrity of the package was evaluated with a hedonic scale: how was that described? How many assessors? Results of this indicator have not been presented. Authors should reconsider shifting this part included in point 2.2.1 (i.e. physical analysis) to point 2.2.5 (sensory analysis).
REPLY: we appreciate the helpful suggestion. Material and Methods were reorganized as suggested. Integrity results were also added to Table 7.
Line 319: The packaging defects (presence of holes/abnormal traces), presence of ice and fillet’s size variation were evaluated by 5 assessors when the packages were opened using a 5-point hedonic scale ranging from defect absent (1 point) to very intense (5 points).
Which was the size of the samples to be cooked? (lines 123-126).
REPLY: we appreciate the request for accuracy. The size of the sample (100 g) presented to the test panel was introduced in the text.
Line 333: For evaluation of cooked fillets, a piece (100g ± 5g) of the larger area from each thawed fillet was individually wrapped in aluminium foil (food use) and steam‐cooked for 10 min at 100 °C in a steam oven (Combi‐Master CM64; Rational, Lund, Sweden) with air circulation, without adding salt or spices.
Please indicate if in the determination of soluble protein the non-protein nitrogen content was subtracted (lines 136-137).
REPLY: we appreciate the request for accuracy. The non-protein nitrogen was not determined. For improved accuracy the information was added in the discussion of the results.
Line 560 : Comparison of the present study data with published levels, shows high soluble protein contents, particularly in the fillets with the longest storage periods. This difference may be due to the contribution of non-protein nitrogen, not accounted in the referenced data because specific protein quantification methods were used (e.g. Lawry and Biuret methods). The NPN-fraction (non-protein nitrogen) has reported to be 16.8% of the total nitrogen in M. productus [84].
No information on the number of packages analysed per brand in each of the indicators is found. Did authors sampled all packages for all brands? Perhaps a sampling plan could help here (a diagram for example).
REPLY: we appreciate the request for clarification. Since the sampling plan was relatively straightforward, the information was improved in section 2.1.1.
Line 191: A total of 13 packages of the same brand and batch were collected in October 2018 in the selected food shops, transported to the laboratory under controlled temperature (-18 oC) and kept stored at -20°C ± 1oC until analysis within 1 week. From these, 10 packages were used for analysis of packaging, net weight, glaze ice, and sensory analysis and 3 packages (pooled) used in the other determinations. Individual weight of hake fillets packages ranged from 0.4 kg to 1.0 kg and no record of the storage temperature was obtained from retailers.
Line 163: Please indicate in which units are results expressed.
REPLY: we appreciate the request for clarification. The information regarding the units of the results was added to the text.
Line 269: Results are expressed as g P2O5/kg in the case of hake fillets and g P2O5/L in the case of thawing waters.
Line 214-216: It is not clear how the overall quality determination was performed. Please clarify.
REPLY: we appreciate the request for accuracy. Text was edited for clarification.
Line 328: Overall quality of raw products was determined as the average of the mean value of each evaluated attribute together with the scores from integrity of package, fillet’s size variation and presence of ice.
Lines 231-237: Please specify which quality parameters were used in PCA.
REPLY: we appreciate the request for accuracy. Text was added for clarification.
Line 349: The parameters included in PCA were storage length, glaze ice, moisture, protein, M/P ratio, soluble protein, soluble/total protein, thaw drip loss, thaw drip loss protein, WHCraw, WHCcook, WHCtotal, TVB-N, pH, total phosphates in fillets, total phosphates in thawing waters, orthophosphates in fillets, orthophosphates in thawing waters, and sensory analysis (overall score of negative descriptors of raw hake fillets, overall score of positive descriptors of cooked hake fillets, and overall score of negative descriptors of cooked fillets).
Line 355: Polyphosphates and citrates contents were not considered for the multivariate analysis on account of the influence of zero data (values lower than LOQ or LOD) for the dispersion of samples points in the plot. Regarding the sensory attributes used in the PCA, in raw fillets, the overall score of negative descriptors was determined by calculating the mean values of the scores of colour (yellow) and odour (rancid). In cooked fillets, the overall score of negative descriptors was determined by calculating the mean values of the scores of discolouration, dehydration, rancid odour, and rancid/bitter flavour, while the overall score of positive descriptors was determined by calculating the mean values of the scores of typical colour, typical odour, typical flavour and texture (firmness and succulence).
Results and discussion
Table 3: The column “net weight difference” does not seem to reflect this. If understood properly, it is calculated as a percentage of actual vs declared weigh. Please rephrase.
REPLY: we appreciate the request for accuracy. The values of “net weight difference” in Table 3 were changed in order to show the difference in terms of percentage of actual vs declared weigh.
Line 273: …”which the need to adjust a specific net weight at packaging does not justify”. The sentence seems weird. Please rephrase.
REPLY: we appreciate the request for improvement. The sentence was rephrased.
Line 402: Nevertheless, the packages fillet’s size variation was relevant, with 79% of the packages showing between 50 and 75% of differently sized fillets (Table 6), which can’t be justified by the need to adjust a specific net weight at packaging.
Lines 275-278: the statement “…showing glazing levels higher than the normal range used in the industry” seems contradictory with authors´ results. Actually they show that the actual levels used in industry range between 5.8-32.9%. It is true that their results contrast with those of data from reference 40. But this may show one part of the industry whereas authors´ data show another part. I suggest to rephrase the paragraph.
REPLY: we appreciate the request for improvement. Own not published data shows that the ice glaze level in 26 frozen hake products of the Portuguese market is lower than 10%. However, as the reviewer mentions it is possible that the industry has changed procedures, surely not in the correct directions as showned by the use of higher than necessary glaze ice levels mentioned by reference 40. The paragraph was rephrased to accommodate the clarification.
Line 410: No maximum or minimum added glaze ice has been regulated for fish and fishery products though addition of 5% to 10 % of the total weight should be sufficient to afford protection in most cases of properly wrapped and sealed products[65]. The values in hake samples ranged between 5.8% and 32.9% (Table 3) with a large majority of the products (79%) showing glazing levels higher than the recommended range.
There are several grammar errors that should be corrected. For example, Line 291: Replace “restores” by “restore”.
REPLY: we appreciate the request for improvement. A revision was made and hopefully all error corrected.
Lines 296-298: The sentence: “Processes occurring during frozen storage, like protein surface dehydration or, to a minor degree, physical damage in the membranes or cells have been considered the causative processes of this decrease” (i.e. water holding capacity). This is an example of descriptions that should described at once and in a more precise way.
REPLY: we appreciate the request for improvement. A revision was made towards transition of this type on information to Introduction.
Lines 299-303: Thaw drip loss of samples 9, 15, and 16 may arise from water addition to Merluccius productus. Rough calculation (i.e. sample number 9) suggest that, for average moistures of 80% for hake, given that moisture of this sample is about 83% (table 4) there was about 90 g of added water in 600 g of product. This would mean that about 15 g out of the 25 g of drip loss may come from added water. I suggest to think around these lines since chances are that rather than having a species-difference in quality, the differences observed in M productus are, in this case, due to this factor. Please note that these are the three samples that have some traces of phosphate and according to authors, they come from the same producer. If this is true, then authors could revise their discussion.
REPLY: we appreciate the call for attention and the suggestion of the reviewer. It is true that traces of tripolyphosphate were found in the ice glaze of samples #9, 15 and 16, not in the muscle, and that this could point to some absorption or improved retention of water during storage. However, tripolyphosphate are also known and used for reducing the drip loss during thawing. It is also known that M. productus is one of the hakes with the higher M/P ratio, (Merluccius capensis/paradoxus 4.7 ± 0.2; Merluccius hubbsi 4.8 ± 0.2; Merluccius productus 5.0 ± 0.2 [85]), so 83% could be a normal moisture content for this species. M. productus is also known for having the softer texture within all the hake species and therefore less appreciated, which could be the reflect of a lower ability to retain water. All factors considered it is not clear the reason for the higher drip loss, but owing to the existence of only trace levels of tripolyphosphates in the ice glaze of the samples and the effect of these on the reduction of drip loss, we fill less comfortable in advancing the effect of tripolyphosphate as the reason for the higher level of drip loss in M. productus. However, if the reviewer thinks that we have here a strong evidence we are open to follow the suggestion.
Lines 401-402: Correlations are mentioned between pH and soluble protein but the latter has not been described yet. Please reorder this type of statements here and throughout the text.
REPLY: we appreciate the remark and the suggestion of the reviewer. Statements regarding correlations between pH/soluble protein, thaw-drip/soluble protein and soluble protein/overall quality evaluation of cooked fillets were reordered.
Lines 435-437: Please consider if the unexpectedly high values authors find are due to non-protein nitrogen. If this is not separated from the protein soluble fraction (i.e. by precipitation of soluble proteins with TCA or PCA for example) this nitrogen may be erroneously considered protein nitrogen.
REPLY: we appreciate the remark. As previously mentioned in the reviewer Material and Methods section the non-protein nitrogen was not determined. For improved accuracy the information was added in the discussion of the results.
Line 560: Comparison of the present study data with published levels, shows high soluble protein contents, particularly in the fillets with the longest storage periods. This difference may be due to the contribution of non-protein nitrogen, not accounted in the referenced data because specific protein quantification methods were used (e.g. Lawry and Biuret methods). The NPN-fraction (non-protein nitrogen) in M. productus has been reported to be 16.8% of the total nitrogen in the muscle [84].
Lines 460-462: The practical storage time in the frozen state increases considerably by lowering the storage temperature. -18 ºC is too high. Authors may want to include this point in the discussion of this part. At least some industries keep their products well below 18 ºC so there may not necessarily by incorrect practices (i.e. mislabelling).
REPLY: we appreciate the remark. We agree that the lower the temperature the better, but according to several references (e.g. Regulation (EC) No 853/2004, Codex Alimentarius, International Institute of Refrigeration) -18°C or lower is the recommended storage temperature for fish and fishery products. A temperature of -18°C is in fact the temperature that many companies use as critical control point in their HACCP plans. Nevertheless, we edited the text to incorporate the possibility of use by the industry and retailers of temperatures lower than -18 ºC. Regarding mislabeling, we know by personal contact with the industry that repackaging of products with expired dates is an incorrect practice used by some producers. Likewise, there are certainly some in the processing and retail industry that use correct procedures (e.g. <-18 ºC) and for this reason we edited the text.
Line 599: Likewise, protective packaging strategies could have an effect of the SP/TP percentage, as well as the use by the industry and retailers of lower than -18 oC storage conditions, but incorrection of labelled freezing dates is also a possibility.
Section 3.5. Total Volatile Nitrogen is mainly used for monitoring microbial changes during storage temperatures above 0 ºC. Because hake is a formaldehyde-forming species, authors could have chosen to analyse TMA and DMA, rather than TVBN. Increase of DMA [and formaldehyde (FA)] values are relevant at -18 ºC but the reaction halts at -30 ºC. It is well known that FA has a detrimental effect resulting in tough products owing to the aggregation of muscular proteins.
REPLY: we appreciate the remark. We agree with the comments and the higher relevance of this parameter at storage temperatures above 0oC, but since TVB-N is regulated for unprocessed fishery products (Commission Regulation (EC) No 2074/2005) and fish fillets fall in this category, the analysis was performed. We also know that analysis of DMA and particularly of FA would have been an added value to the study, but due to laboratory limitation that was not possible.
Line 625: Authors state that there is no microbiological problem since hake fillets are eaten after thoroughly cooking. This is an opinion. They should state if this is within the accepted levels, since they mention that the presence of moulds reflects poor hygienic conditions. A table with the microbiological data per brand would be of help.
REPLY: we appreciate the comment. The text was edited and information regarding the regulatory framework in terms of microbiology limits was added in the Introduction. Also, in order to avoid misinterpretations, the final sentence was deleted and added information regarding the nonexistence of established limits for moulds and yeasts for frozen fish and a table with the results.
Line 757: Though there are no established limits for moulds and yeasts for frozen fish, the presence of levels up to 2 log cfu/g of yeasts and moulds in six samples indicate poor hygienic conditions in some of the production facilities during hake handling before freezing.
Line 781: Table 6 – Counts (cfu/g) and presence (Salmonella) of microorganisms in frozen pre-packaged hake fillets sampled in the Portuguese retail market (n=3).
Lines 640-642: Hake suffers mainly aggregation.
REPLY: we appreciate the call for precision. The text was edited.
Line 52: Although freezing of meat is an effective method of long-term preservation, fish and fishery products can suffer undesirable changes during frozen storage (e.g. protein denaturation and aggregation, enzymatic hydrolysis of lipids and proteins) and deterioration limits the storage time.
Line 79: Induced by denaturation/aggregation of proteins and quality deterioration throughout frozen storage, cook loss occurs also during the heat processing of hake fillets.
The actual practical storage time depends, all the rest of factors being constant, on the time but also the temperature of storage. Thus if samples have been stored at -25 ºC (some industries do this), two years may be adequate. I suggest to consider this in the discussion.
REPLY: we appreciate the comment. We acknowledge that some industries may store frozen fishery products at -25 ºC and that in fact the lower the storage temperature, the longer the best before period. However, the general low sensory quality of the products does not support the use of these lower storage temperatures. Of course there are a few exceptions of products of acceptable quality, probably from those that use lower temperature storage.
I suggest to include tables with sensory analysis of raw and cooked hake fillets with information by brand, as it was done in the former analyses (i.e. tables 1-5).
REPLY: we appreciate the suggestion. The authors considered that by presenting the data in this way, instead of a table with all the individual scores of each sample, the final results expressed in terms of number of times the different sensory scores occur in the sensory evaluation of the 19 commercial hake fillet products, are more informative. In this format it is possible a better evaluation of the overall performance of the brands in regard to the different attributes. We were more interested in the overall performance of the industry and retailers rather than the individual evaluation per brands.
In table 6 it is not clear how the overall quality has been obtained. It seems strange that if there is 5% of the brands (i.e. one out of 20) displaying score 5 (Bad) in 5 out of 7 descriptors, the overall quality for a bad sample is 0%. Of course, chances are that there is not always the same brand having these bad scores, but it should be better explained.
REPLY: we thank the comment. The reviewer is correct, there is no overall bad quality score (5) of raw hake fillet because these scores (5) are not all in the same sample and also because the descriptors Package Defects, Size Variation and Presence of Blood Stains showed relatively good levels and contributed in the mean evaluation for a value lower than 5. Text was edited in order to clarify.
Line 319: The packaging defects (presence of holes/abnormal traces), presence of ice and fillet’s size variation were evaluated by 5 assessors when the packages were opened using a 5-point hedonic scale ranging from defect absent (1 point) to very in-tense (5 points).
Line 328: Overall quality of raw products was determined as the average of the mean value of each evaluated sensory attribute together with the scores from packaging defects, fillet’s size variation and presence of ice.
Table 7 is difficult to follow since the positive and negative scores have the same scale (i.e. absent=1) but meaning completely different things, and are on the same row, and thus it is difficult to get an idea of what is happening. Moreover, Overall quality (i.e. 1, bad) is on the same row of rancid (also score 1, absent). I suggest to change the laying out of the table and make it easier to read. Again, this reviewer suggest to include tables with the actual values of each descriptor per brand. This would help the reader relate the biochemical with sensory values more easily.
REPLY: we appreciate the reviewer comment. We understand that the reading of the table may be difficult if there is no acknowledgement and differentiation by the reader between the positive and negative attributes. For better understanding of the table we changed the layout and identified in the legend the positive and negative attributes. Since a hedonic scale of intensities of positive and negative attributes is used a table with the actual values of each descriptor per brand would have the same configuration and was therefore not the preferred choice.
Line 828 and 862: Table 7 and 8
What does firmness (1-absent) mean?
REPLY: we thank the request for clarification. Absent firmness (1) is the lower level of the texture scale and was considered equivalent to absence of cohesion, absence of toughness, soft or mushy texture.
Figure 1, replace PCA with PC.
REPLY: we appreciate the call for correction. PCA was replaced with PC both in figure 1 and in the text, accordingly to reviewer suggestion.
Line 890: Figure 1
Line 706: lower PCA1 values is meaningless. Please rephrase thoughout the text.
REPLY: we appreciate the remark and the suggestion of the reviewer. The text was edited according to the reviewer proposal.
Line 709: from this statement, it looks that PCA has been performed also with sensory values. Please explain which variables were used in the analysis.
REPLY: we appreciate the request for clarification. Yes, PCA was also done with sensory analysis results. The variables overall score of negative attributes of raw fillets, overall score of negative attributes of cooked fillets, and overall score of positive attributes of cooked fillets were included in the principal component analysis.
Please note that results from fig1 may not be species related. As explained before, they may be due to the fact that samples 9, 15 and 16 are from the same processor and they all (may) have added water. (Fig 1 and lines 721-726).
REPLY: we appreciate the call for attention. In the graph, different species were identified with different symbols for an easier visualization, taking into account that several parameters were previously associated with the species. Separation of samples in the graph may not be species related, but there are some characteristics of the species that have contributed for the separation of samples in the graph, namely the texture of fillets which is reflected e.g. in thaw drip loss and in sensory evaluation. Added phosphates (water addition) also contributed for the separation of samples 9, 15 and 16 (all of M. productus). Also, as previously referred, addition of water is less likely to have happened in M. productus samples and therefore not accounted as a contributing factor to separation of species.
As regards factor analysis, please consider including a loading table, so that readers can see which are the variables that entered in each of the PC. Also, please indicate which is the % of explained variance for each PC.
REPLY: we appreciate the reviewer suggestion. In the text it is included a paragraph (lines 895-902) that relates the first three PCs with the variables for which the higher loadings were obtained. Following the reviewer suggestion, a table with the loadings was added to the manuscript (Table 7). The % of variance explained for each PC is indicated in the legend of figure 1 “PC 1, PC 2, and PC 3 explained 44.5%, 19.2%, and 9.8%, respectively, of the variation of the original variables.”
Line 903: Table 9
Conclusions
As stated before, the conclusions are in some aspects based on opinions rather than in facts. For example, in lines 775-776, one may argue on what bases the quality has been defined as poor to medium. I suggest to be more specific and differentiate between what is mandatory by regulation (criteria and threshold levels), in which criteria there is a wide scientific consensus for a given threshold, and what it is a suggestion of the authors.
REPLY: we appreciate the reviewer remarks. The text was edited in order to accommodate the points mentioned.
Occasionally, results are presented in the conclusions without being mentioned before. For example, in lines 777-779, authors mention that fillets from M. paradoxus were the most appreciated by the sensory panellists…etc but this was not presented before.
REPLY: we appreciate the reviewer comment. The preference for M. paradoxus was derived by the evaluation of the positive descriptors, namely texture, as mentioned in the multivariate analysis section. However, taking into account that the information did not add much to the conclusion the text was removed.
Lines 780-784, please see the previous comment on the cooked samples/microbial analyses.
REPLY: we appreciate the reviewer comment. The text was edited in order to clarify the results.
Line 959: In relation to the microbiological contamination pathogens were absent and though some hygiene indicators were low, the presence of coliform, E. coli, moulds and yeasts in some samples indicate poor hygienic conditions in a number of production facilities during hake fillets handling before freezing. Nevertheless, the global microbiological quality of hake fillets was good and products do not present any significant health safety issues.
Line 791: ‘best before’ longer than recommended… by whom? I strongly suggest to look carefully at the conclusions.
REPLY: we appreciate the reviewer call for attention. Several recommendations have been published regarding the practical storage life of lean fish at -18 oC, namely 8 months [25], 9 months [9] and between 8-10 months [26]. Of course, it is possible that some producers or retailers use lower temperatures, which would naturally extend the best before date, however the general low sensory quality of the products does not support the use of these lower storage temperatures.
Line 794: The sentence: “No product has been rated as good or excellent by a trained sensory panel”. Hedonic scales are usually done for consumer studies but with a much higher number of persons. Chances are that results are biased so that they not reflect the view of the average consumer.
REPLY: we appreciated the reviewer comment. We accept that the evaluation of a trained sensory panel may be more demanding and meticulous than that of an ordinary consumer and therefore possibly biased in that sense. However, it was specifically for this reason that a trained panel was used. It was the objective of this study to have an evaluation, as professional and accurate as possible of the quality of the frozen hake fillets available in the Portuguese market. For this reason, the high level of expertise of the assessors together with a hedonic scale that had the advantage of giving judges more freedom to express their sensory perceptions, was the authors option.
I suggest Allan Bremner´s paper (Toward Practical Definitions of Quality for Food Science, 2000, Critical Reviews in Food Science and Nutrition 40(1):83-90). He defines an approach which links the concept of quality, through a general definition, by adding the “missing link of specific definitions related to measurable attributes and properties determined by standard methods to provide values that can be used to evaluate foods or to set specifications”.
REPLY: we appreciated the helpful insight of the reviewer. The authors incorporated the concept in the text.
Line 173: Following the approach of Bremmer [40] that linked “the concept of quality, through a general definition, by adding the missing link of specific definitions related to measurable attributes and properties determined by standard methods to provide values that can be used to evaluate foods or to set specifications” a multivariate analysis of all data was performed to identify the most informative quality control parameters and also to detect groups of commercial hake samples with similar quality characteristics.
Round 2
Reviewer 2 Report
Authors have responded to all the issues addressed by this reviewer in a satisfactory way. I congratulate them for the significant improvement of the manuscript and I consider the paper acceptable.